# Plan-Answer-Refine-on-Graph: Structured Planning and Self-Refinement for Large Language Model Reasoning on Knowledge Graphs

**Yuxin Shi[1], Han Fu[1,3†], Zhuo Li[2], Chenghao Liu[4*], Xiaoxue Ren[1†], Jianling Sun[1]**
[1] Zhejiang University, [2] State Street Technology (Zhejiang) Ltd, [3] Richoo AI, [4] Datadog AI Research
{yxshi,lizhuo,xxren,sunjl}@zju.edu.cn   fuhan@richoo.ai
twinsken@gmail.com

## Abstract

Incorporating knowledge graphs (KGs) into large language model (LLM) reasoning has shown promise in alleviating hallucinations and factual errors. Although existing paradigms of KG-augmented LLMs have achieved encouraging results, they still exhibit notable limitations when handling multi-hop reasoning and complex logical queries: (1) search space truncation bias: current methods generate linear entity-relation reasoning paths, which can prune correct candidates prematurely during iterative exploration; and (2) entity error amplification: existing methods typically follow the retrieve-and-answer paradigm which causes LLMs to over-rely on retrieved evidence, exacerbating the impact of incorrect entities during reasoning. To alleviate the existing challenges, we propose Plan-Answer-Refine-on-Graph (PARoG), a novel framework for LLM reasoning on knowledge graphs. First, PARoG leverages SPARQL queries from KG data as references, decomposing them into structured step-by-step plans. We further train LLMs to construct such structured plans, which improves the logical consistency of reasoning, ensures uniform step granularity, and facilitates effective execution on the graph. Second, during reasoning over KGs, PARoG adopts a plan-answer-refine paradigm: the model first attempts to answer each sub-query independently, and then refines its prediction by integrating evidence retrieved from the KG. This process mitigates knowledge conflicts between LLM and KG, substantially reducing hallucinations. Experimental results on multiple KG reasoning benchmarks demonstrate that PARoG significantly outperforms state-of-the-art approaches, achieving especially superior accuracy on multi-hop and logically complex queries. Our code is available at https://github.com/shiyuxin2000/PARoG

## 1 Introduction

Large Language Models (LLMs) (Brown et al., 2020; Ouyang et al., 2022; OpenAI et al., 2023; Dubey et al., 2024; Guo et al., 2025) have demonstrated remarkable reasoning capabilities in a wide range of complex natural language processing tasks (Bang et al., 2023; Zhao et al., 2023; Huang & Chang, 2023; Qiao et al., 2023). However, LLMs remain prone to hallucinations and factual errors in real-world applications due to their reliance on implicit parametric knowledge (Hu et al., 2023; Wang et al., 2023a; Huang et al., 2024). Knowledge graphs (KGs), as large-scale structured external source of factual knowledge, offer explicit, interpertable relational structures which can ground LLM reasoning, providing a natural complement to limitations of LLMs (Pan et al., 2024).

Recent LLM⊗KG approaches can be categorized into two paradigms. The first leverages step-wise graph exploration, where LLMs iteratively perform entity–relation walks to progressively construct reasoning paths (Sun et al., 2024; Ma et al., 2025). The second generates global reasoning plans where questions are decomposed into sub-objectives and the KG is queried along the planned path

---

*Work was done before joined Datadog
†Corresponding authors.

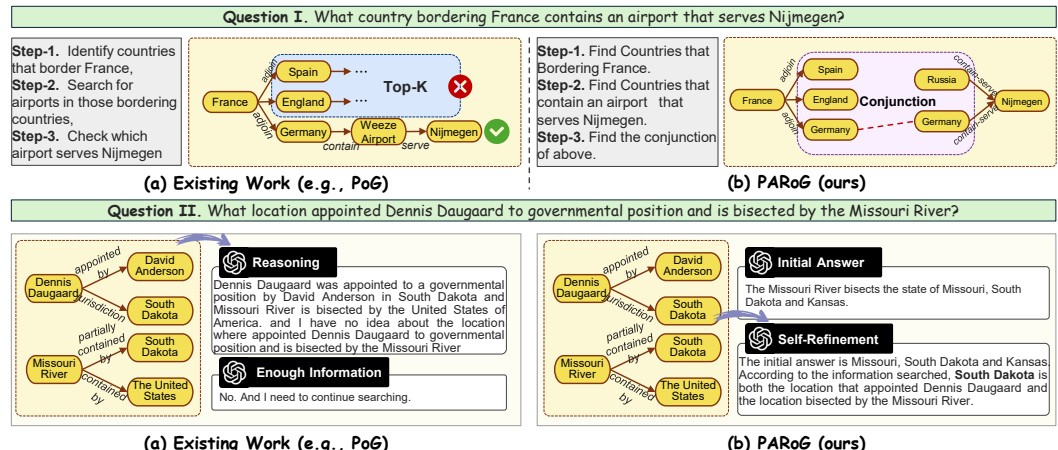

Figure 1: Illustration of the challenges in the existing methods and how our proposed PARoG addresses these issues: I. Search Space Truncation Bias and II. Error Amplification.

to obtain external information (Luo et al., 2024; Chen et al., 2024b). Though demonstrating notable improvements, these methods often struggle with complex logical queries that involve conjunctions or multiple constraints. Our systematic analysis of existing approaches (as described in Appendix B) identifies the following two fundamental limitations.

**Search Space Truncation Bias due to Linear Reasoning Paths.** Current methods construct reasoning paths primarily along linear entity–relation steps, iteratively expanding from one entity to its neighbors. To control the combinatorial explosion of graph exploration, they prune candidate entities at each step (*e.g.*, using top-k selection). While efficiency, this strategy often eliminates correct entities prematurely. For instance in Figure 1 (I-a), the correct answer *Germany* is eliminated early due to pruning, leading to an incorrect prediction. A more reasonable planning strategy would first decompose the question into two sub-problems: (i) identify countries bordering France, and (ii) identify countries with airports serving Nijmegen, and then compute the conjunction of these results. The limited planning capability of existing methods fundamentally biases the search space and limits reasoning performance.

**Error Amplification from Faulty Entities and Relations.** LLM-generated reasoning paths may introduce spurious or weakly related entities and relations during KG exploration. Existing methods typically follow a retrieve-and-answer paradigm, where the LLM heavily relies on the retrieved evidence to produce the final answer. This reliance amplifies errors. For example in Figure 1 (II-a), during graph-based reasoning, the system retrieves facts such as *"Dennis Daugaard was appointed by David Anderson in South Dakota"* and *"Missouri River is partially constrained by South Dakota, USA"*. Though individually correct, the knowledge are not sufficiently directly connected to answer the question. Existing methods typically make the LLM to over-rely on the retrieved information, attempt further reasoning steps, and ultimately fail to produce the correct answer.

To alleviate theses challenges, we propose a Plan–Answer–Refine framework (PARoG), a hybrid reasoning paradigm that tightly integrates structured explicit guidance with parametric LLM reasoning. As shown in Figure 2, our method introduces two key technical contributions. First, we leverage SPARQL queries as the structured references to supervise planning and train the planning module using a relatively smaller model (*e.g.* Llama-3.1-8B) to generate flexible, compositional reasoning paths that allow complex logical operations over sub-queries (*e.g.* conjunctions, compositions, superlatives and comparatives). For example in Figure 1 (I-b), instead of searching sequentially from "France" to its neighboring countries and then their airports, the model can generate conjunctive sub-objectives such as "find countries bordering France" and "find countries with airports serving Nijmegen," then reason over the combination of the sub-objectives, which mitigates search space truncation bias by moving beyond linear expansions. Second, rather than committing to retrieved entities in a one-shot retrieve-and-answer paradigm, PARoG first produces a tentative answer using

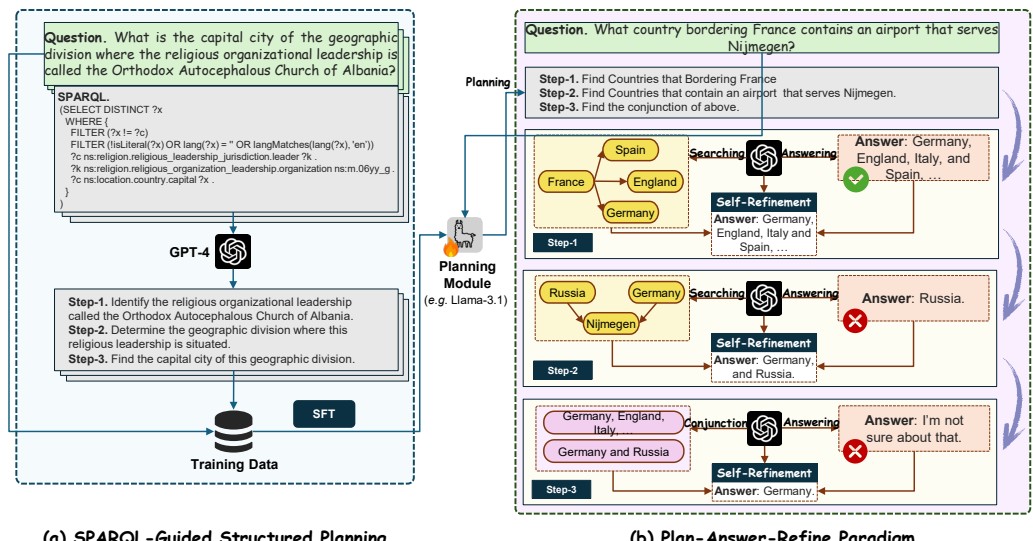

Figure 2: Overall framework of the proposed PARoG. Unlike prior methods that sequentially expand entity–relation paths with pruning, and follow the one-shot retrieve-and-answer paradigm, PARoG combines (a) structured planning with (b) iterative self-refinement, enabling robust handling of complex logical queries with conjunctions, compositions, comparisons, superlatives.

its parametric knowledge and then explicitly refines it by referring to the retrieved KG entities as shown in Figure 1 (II-b). This refinement step overrides earlier faulty evidences, preventing error amplification due to spurious entities or weakly related relations. Our main contributions are as follows:

- We propose leveraging SPARQL as structured references to supervise planning to train the model to generate compositional reasoning paths, which enables the model to handle complex logical reasoning including conjunction, composition, superlative and comparative queries.

- We propose a plan-answer-refine framework, where the agent first attempts to answer then explicitly refines the results using retrieved evidence. This step reduces error propagation caused by faulty entities or relations involved in the reasoning paths.

- We introduce a novel framework PARoG by combining the proposed techniques and evaluate the performance on multiple real-world KGQA datasets. The experimental results demonstrate significant improvements over state-of-the-art baselines.

- We provide further analysis demonstrating that PARoG uses a relatively small model (*e.g.* Llama-3.1-8B (Dubey et al., 2024)) to generate reasoning paths, yet its performance can surpass larger planning LLMs (*e.g.* ChatGPT OpenAI et al. (2023) or Deepseek-R1 (Guo et al., 2025)). We also discuss the broader impact of using structured symbolic guidance with LLM reasoning beyond KGQA.

## 2 PRELIMINARIES

**Knowledge Graph.** A *knowledge graph* (KG) is composed of a large set of fact triples, represented as a graph $\mathcal{G} = \{\langle e, r, e' \rangle \mid e, e' \in \mathcal{E}, r \in \mathcal{R}\}$, where $\mathcal{E}$ and $\mathcal{R}$ denote the sets of entities and relations respectively. For KGQA tasks in this paper, we assume the availability of a KG that contains the entities relevant to answering the given natural language question.

**SPARQL Query Language.** *SPARQL* (SPARQL Protocol and RDF Query Language) is a formal query language which allows users to query structured knowledge bases. Given a question $q$, the SPARQL query $\mathcal{S}$ specifies a pattern of triples to match against the knowledge graph $G$. A general

SPARQL query consists of a `SELECT` clause that specifies the variables to retrieve, and a `WHERE` clause that defines the graph pattern to match. In our method, SPARQL queries are used to supervise the generation of reasoning paths by decomposing complex queries into smaller sub-queries. Figure 2 provides an example of SPARQL query.

## 3 METHODOLOGY

In this section, we present PARoG, a novel hybrid reasoning framework which integrates structured guidance planning with parametric reasoning and refinement over knowledge graphs. As shown in Figure 2, PARoG comprises 2 major stages:SPARQL-Guided Structured Planning-training LLMs to generate compositional planning of sub-objective paths based on SPARQL-guided supervision for knowledge graph exploration, and Plan-Answer-Refine Paradigm-iteratively completing a sub-objective with parametric knowledge and then correcting the answer using external KG evidence to mitigate errors and inconsistencies.

### 3.1 SPARQL-GUIDED STRUCTURED PLANNING

The current LLM⊗KG paradigm typically generates linear entity-relation reasoning paths. In this process, the model starts from an entity and iteratively explores its neighbors by following predefined ⟨*entity-relationship-entity*⟩ paths . While effective for simple queries, this linear path generation often fails to capture complex multi-step reasoning required for more sophisticated queries. Specifically, when reasoning over complex questions involving compositionality, conjunctions, comparatives, or superlatives, LLMs inherent behavior of linear exploration can lead to *Search Space Truncation Bias*, where pruning intermediate candidates prematurely eliminates correct answers.

**SPARQL-Guided Supervision.** To address this issue, we propose to use SPARQL queries as a structured guide for reasoning. SPARQL inherently supports complex queries that involve logical operations. In this paper, we consider the following operation types:

- **Conjunctions**: finding entities that satisfy multiple constraints simultaneously. *e.g. Find countries that border France and have airports serving Nijmegen.*

- **Compositions**: expressing queries where the output of one relation serves as the input to another. *e.g. Find the capital city of the country that has airports serving Nijmegen.*

- **Comparatives**: retrieving entities based on relative attributes. *e.g. Find countries larger than France in area and have airports serving Nijmegen.*

- **Superlatives**: selecting the best entity according to a ranking predicate. *e.g. Find the largest city bordering France.*

When the intermediate candidate set is large, these query types cannot be properly handled using linear entity-relation paths but are essential for real-world KGQA tasks.

**SPARQL-to-Planning.** To transfer this expressiveness into model training, we leverage SPARQL queries as guidance signals and use state-of-the-art LLMs to automatically generate planning data from complex questions. For each input question, GPT-4o produces a set of decomposed sub-questions that reflect the logical structure of the underlying SPARQL query. Specifically, we design a systematic pipeline to automatically construct a large-scale dataset tailored for KGQA tasks. The graph-matching process in SPARQL naturally decomposes a complex query into a sequence of consecutive search operations and constraints, thereby providing a precise planning path for identifying intermediate sub-objectives. Building upon this observation, the SPARQL-to-Planning pipeline consists of the following two steps.

- **Source Data Collection.** We first select diverse ⟨*Question*, *SPARQL*⟩ pairs of multi-hop queries from public KGQA training datasets including WebQSP(Yih et al., 2016), CWQ (Talmor & Berant, 2018), and GrailQA (Gu et al., 2021). These pairs serve as the foundation for aligning natural language with structured reasoning.

- **Semantic Consistency Mapping.** With the collected data, we decompose the SPARQL queries into sub-operations and then translate each atomic operation into a fluent natural language question as single sub-objective of the reasoning plan. Following that, we also rephrase the decom-

posed sub-objective sequence back to natural language queries to maintain the semantic consistency. Instead of the original questions, we use the rephrased natural language queries and the generated sub-objectives as the training data

During dataset construction, the SPARQL queries are decomposed into atomic operations to maintain the consistency across plan steps. In this paper, we use GPT-4o Hurst et al. (2024) to automate the overall process. Finally, the pipeline produces 74,802 high-quality decomposition examples covering a wide range of query types and reasoning depths. The statistics of different query types are summarized in Figure 3.

**Model Training.** We employ a relatively small but powerful open-source model Llama-3.1-8B Dubey et al. (2024) as the foundation backbone. The training objective follows the standard autoregressive language modeling loss. We provide the full input template in Appendix C. Given the input tokens $\mathbf{x}$, the model parameters $\theta$ are optimized by minimizing the negative log-likelihood:

$$\arg \min_{\theta} \mathcal{L}(\theta) = -\sum_{i}^{H} \sum_{j}^{T_h} \log P_{\theta}(o_{i,j} | \mathbf{o}_{i,<j}^{h}, \mathbf{x}) \tag{1}$$

where $H$ and $T_h$ deba the total number of sub-objectives and the token number of a single sub-objective respectively, and $\mathbf{o}_i = \{o_{i,1}, \cdots, o_{i,T_h}\}$ is the $h$-th sub-objective. With supervised training, the model learns to map complex natural language questions into sequences of structured sub-questions which mirror SPARQL compositional logic. This training equips the planning module with the ability to produce complex logical reasoning paths (*e.g.* conjunctions or comparatives), ensuring correct entities are preserved during exploration and mitigating the Search Space Truncation Bias of existing approaches.

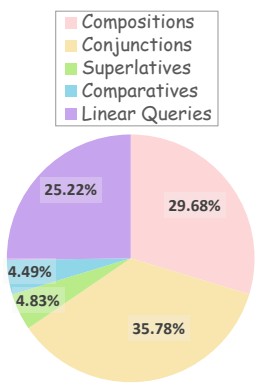

Figure 3: Statistics of different query types of the generated planning data.

## 3.2 PLAN-ANSWER-REFINE PARADIGM

Another fundamental challenge for LLM⊗KG reasoning is *Error Amplification from Faulty Entities and Relations*. Existing methods typically adopt the one-shot retrieve-and-answer paradigm, where LLM generates a reasoning path, retrieves corresponding entities and relations from the KG, and then directly uses these retrieved facts to finalize an answer. Though intuitive, this paradigm suffers from two issues:

- **Error Propagation.** When a spurious or weakly related entity is introduced by the reasoning path, the subsequent steps will propagate and accumulate this error.

- **Over-Reliance on Retrieval.** LLMs often assumes the retrieved evidences from KG to be correct and sufficient, even when the external information only partially address the query. This over-reliance prevents the model from self-correcting, leading to faulty answers.

To mitigate this challenge, we introduce the plan-answer-refine mechanism. We employ the LLMs to generate a tentative answer using the parametric reasoning ability and then leverage the KG reasoning agent to iteratively explore the knowledge graph to obtain external information and refine the answer by adjusting entities or relations. The algorithmic procedure of this mechanism is summarized in Appendix D.

**Answering.** Let $\mathcal{O}$ denotes the reasoning plan generated by the planning module. For each sub-objective $o_i \in \mathcal{O}$ generated by the planning module, the initial step is leveraging a LLM $\mathcal{M}$ to generate a tentative answer as:

$$\hat{a}_i = \mathcal{M}(Q, o_i, I_A) \tag{2}$$

where $I_A$ denotes a predefined instruction template and $Q$ is the input question.

**Exploration.** The KG exploration process of PARoG is similar to existing work (Sun et al., 2024; Chen et al., 2024b). To be specific for each sub-objective, PARoG starts from an initial entity and iteratively exploring the knowledge graph. Following previous work (Sun et al., 2024; Jiang et al., 2023a), the iterations begins with a set of $n_0$ topic entities $\mathcal{E}_0 =$

$\{e_1^0, e_2^0, \ldots, e_{n_0}^0\}$. For the $i$-th iteration ($i > 1$), we first obtain the current set of $K$ reasoning paths $\mathcal{P}_{i-1} = p_1^{i-1}, \cdots, p_K^{i-1}$ after previous $i - 1$ iterations. Here, each reasoning path $p_k^{i-1} = [(e_{s,k}^{i-1,1}, r_k^{i-1,1}, e_{o,k}^{i-1,1}), \cdots, (e_{s,k}^t, r_k^{i-1,t}, e_{o,k}^{i-1,t}), \cdots, (e_{s,k}^{T_k}, r_k^{i-1,T_k}, e_{o,k}^{T_k})]$ is a sequence of $T_k$ triples ($T_k < i$) where $t$ indexes the elements, $e_{s,k}^{i-1,t}$ and $e_{o,k}^{i-1,t}$ denote the subject and object entities respectively, and $r_k^{i-1,t}$ is a relation linking them. Then, we continue to extend the reasoning paths forward based on the current triples. Concretely, the set of tail nodes in the current reasoning paths is denoted by $\mathcal{E}_{i-1} = \{e_1^{i-1}, e_2^{i-1}, \cdots e_{n_k}^{i-1}\}$ and the relation set is represented as $\mathcal{R}_{i-1} = \{r_1^{i-1}, r_2^{i-1}, \cdots r_{n_k}^{i-1}\}$. We then expand the reasoning path through searching over relations entities. With the original question $Q$ and sub-objectives $\mathcal{O}$, we leverage the LLMs to select the most relevant relations and entities. Specifically, during the relation searching stage, we begin with all relations connected to the tail entities in $E^{i-1}$, which are denoted by $\mathcal{R}_{init}^K = \{r_{init,1}^i, r_{init,2}^i, \ldots, r_{init,n}^i\}$. and employ the LLM to filter out irrelevant relations. In this step, the entire reasoning plan $O$ is also provided to the LLM so that the model maintains awareness of the global reasoning objective, thereby preventing it from over-focusing on local. Given the tail entities and filtered relations, the missing entities are obtained using predefined SPARQL query templates such as $(e, r, ?)$ or $(?, r, e)$. When all the entities are obtained, we leverage the model to further calculate the relevance between the retrieved entities and the current sub-objective $o_i$ and the question $Q$. The most relevant entities from a large set of candidates are reserved to update the reasoning path set, which is denoted by $\mathcal{P}_i$.

**Self-Refinement.** After each KG exploration iteration, PARoG explicitly re-evaluate the tentative answer against the retrieved evidences. If inconsistencies or supplementary are detected, PARoG refines the result by adjusting entities or relations, effectively correcting errors from earlier steps. Specifically we use the LLM $\mathcal{M}$ to correct answer $a_i$ as:

$$a_i = \mathcal{M}(\mathcal{P}_i, o_i, \hat{a}_i, I_R) \tag{3}$$

where $\mathcal{P}_i$ denotes the set of retrieved triples in the current iteration, and $I_R$ is the instruction prompt. It is also worth noting that we also explicitly ask the LLM to judge whether the retrieved knowledge aligns with the question; if it does not, the generated tentative answer is directly used instead. After each round of self-refinement, PARoG is leveraged to determine whether the current result $a_i$ is sufficient to answer the overall question Q. If the answer is "yes", we stop searching and use $a_i$ as the final answer to avoid over-exploration. Otherwise, PARoG continues iterative searches until PARoG finds enough knowledge or reach the maximum number of iterations. Unlike existing methods, PARoG introduces a mechanism that explicitly integrates the parametric knowledge of LLMs with external knowledge, reducing reliance on any single retrieval and providing resilience against misleading entities or relations.

## 4 EXPERIMENTS

**Datasets.** We conduct comprehensive experiments on multiple Knowledge Graph Question Answering (KGQA) benchmark datasets to evaluate the effectiveness of our proposed approach. Specifically, we utilize three widely-adopted datasets: WebQSP (Web Questions Semantic Parses) (Yih et al., 2016), GrailQA (Strongly Generalizable Question Answering) (Gu et al., 2021), and CWQ (ComplexWebQuestions) (Talmor & Berant, 2018). All three datasets are grounded on the Freebase knowledge graph, which contains 88 million entities, 20K relations and 126 million triplets, making it one of the most comprehensive knowledge bases for KGQA evaluation.

**Metrics.** For evaluation, we adopt the Exact Match accuracy (Hits@1) as our primary metric, which measures the percentage for which the predicted answer exactly matches the ground truth. This ensures that our evaluation strictly reflects the capability to provide precise answers rather than partially correct responses. The results are averaged over three seeds and reported as mean ± 95% confidence interval.

**Compared Methods.** We compare PARoG with 17 LLM-based approaches from 3 categories: (1) LLM prompting methods, (2) LLM reasoning over KGs (LLM $\otimes$ KG), (3) end-to-end fine-tuned KG-augmented LLMs, and (4) graph-retrieval methods. The details of the compared approaches are described in Appendix F.

Table 1: Performance comparison of different methods on two KGQA benchmarks.

| Methods | WebQSP | CWQ |
|---|---|---|
| *LLM Prompting* | | |
| IO (Brown et al., 2020) | 63.3 | 37.6 |
| CoT (Wei et al., 2022) | 62.2 | 38.8 |
| SC (Wang et al., 2023c) | 61.1 | 45.4 |
| *Graph-Retrieval Methods* | | |
| GNN-Rag (Mavromatis & Karypis, 2025) | 82.8 | 62.8 |
| SubgraphRag + GPT4o (Li et al., 2025) | 87.1 | 54.9 |
| *LLM ⊗ KG with GPT-3.5* | | |
| ToG (Sun et al., 2024) | 76.2 | 57.1 |
| RoG (Luo et al., 2024) | 81.5 | 52.6 |
| KG-Agent (Jiang et al., 2025) | 79.2 | 56.1 |
| StructGPT (Jiang et al., 2023a) | 75.2 | 55.2 |
| PoG (Chen et al., 2024b) | 82.0 | 63.2 |
| ReKnowS (Wang et al., 2025) | 81.1 | 58.5 |
| **PARoG** | **89.0** (± 1.3) | **73.1** (± 0.9) |
| *LLM ⊗ KG with GPT-4* | | |
| ToG (Sun et al., 2024) | 80.7 | 65.4 |
| KG-Agent (Jiang et al., 2025) | 81.2 | 67.0 |
| StructGPT (Jiang et al., 2023a) | 79.5 | 64.7 |
| PoG (Chen et al., 2024b) | 87.3 | 75.0 |
| ReKnowS (Wang et al., 2025) | 83.8 | 66.8 |
| **PARoG** | **91.2** (± 0.9) | **79.3** (± 1.1) |

**Implementations.** For SPARQL-Guided Supervision, we use the training split of WebQSP, GrailQA, and CWQ as the source and employ GPT-4 to generate the training data, and the statistics is summarized in Appendix E. We use Llama-3.1-8B as the backbone to train the planning module with learning rate 2e-5 on 4 Nvidia A800 GPUs. We use GPT-3.5 or GPT-4 to serve as the underlying LLMs and report the results on both, thereby analyzing our method across diverse settings.

## 4.1 PERFORMANCE COMPARISON

**Main Results.** The comparison results on WebQSP and CWQ in Table 2. Across both benchmarks, our propose method PARoG consistently outperforms existing approaches. Compared with the state-of-the-art baseline Planning-on-Graph (PoG), PARoG gains substantial improvements of 3.9 and 4.3 points on WebQSP and CWQ respectively with GPT-4. Under the more challenging setting with GPT-3.5, more significant improvements can be observed: PARoG surpasses the baseline by 7.3 on WebQSP and 10.1 on CWQ. It is worth noting that the improvements are particularly pronounced on CWQ, which contains more complex multi-hop and compositional queries, underscoring the advantage of our structured plan-

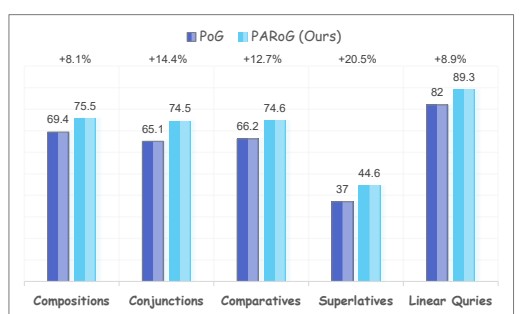

Figure 4: Performance comparison over different query types.

ning and self-refinement mechanism. On the GrailQA benchmark (Table 2), PARoG also achieves consistent state-of-the-art performance across all evaluation settings. Using GPT-3.5, PARoG reaches an overall accuracy of 82.7, surpassing the state-of-the-art Debate-on-Graph (DoG) by a large margin of 4.9 points. Under the stronger GPT-4 setting, our method further improves to an overall 87.2 points, exceeding the compared methods by 2.5 points. It can be observed that the improvements of PARoG are particularly significant on compositional and zero-shot queries, demon-

Table 2: Performance comparison of different methods on GrailQA.

| Method | Overall | I.I.D. | Compositional | Zero-shot |
|---|---|---|---|---|
| *LLM Prompting* | | | | |
| IO Prompt (Brown et al., 2020) | 29.4 | – | – | – |
| CoT (Wei et al., 2022) | 28.1 | – | – | – |
| Self-Consistency (Wang et al., 2023c) | 29.6 | – | – | – |
| *End-to-End Fine-Tuned KG-Augmented LLMs* | | | | |
| RnG-KBQA (Ye et al., 2022) | 68.8 | 86.2 | 63.8 | 63.0 |
| TIARA (Shu et al., 2022) | 73.0 | 87.8 | 69.2 | 68.0 |
| FC-KBQA (Zhang et al., 2023) | 73.2 | 88.5 | 70.0 | 67.6 |
| Pangu (Gu et al., 2023) | 75.4 | 84.4 | 74.6 | 71.6 |
| FlexKBQA (Li et al., 2024b) | 62.8 | 71.3 | 59.1 | 60.6 |
| GAIN (Shu & Yu, 2024) | 76.3 | 88.5 | 73.7 | 71.8 |
| KG-Agent (Jiang et al., 2025) | 86.1 | 92.0 | 80.0 | 86.3 |
| *LLM ⊗ KG with GPT-3.5* | | | | |
| KB-BINDER (Li et al., 2023a) | 53.2 | 72.5 | 51.8 | 45.0 |
| ToG (Sun et al., 2024) | 68.7 | 70.1 | 56.1 | 72.7 |
| PoG (Chen et al., 2024b) | 76.5 | 76.3 | 62.1 | 81.7 |
| DoG (Ma et al., 2025) | 77.8 | – | – | – |
| **PARoG** | **82.7** (± 1.5) | **85.4** (± 0.9) | **66.7** (± 2.0) | **87.1** (± 2.2) |
| *LLM ⊗ KG with GPT-4* | | | | |
| ToG (Sun et al., 2024) | 81.4 | 79.4 | 67.3 | 86.5 |
| PoG (Chen et al., 2024b) | 84.7 | 87.9 | 69.7 | 88.6 |
| DoG (Ma et al., 2025) | 80.0 | – | – | – |
| **PARoG** | **87.1** (± 1.3) | **89.5** (± 2.1) | **73.2** (± 1.9) | **91.1** (± 2.3) |

strating robustness in both complex reasoning and out-of-distribution scenarios. These results highlight that PARoG not only advances the overall accuracy but also generalizes better to complex and zero-shot queries, demonstrating the effectiveness of the proposed methodology.

**Analysis on Different Query Types.** We summarize the comparison between our method PARoG and PoG on different query types in Figure 4. Overall, our method PARoG consistently outperforms PoG across all types. Compared with simple categories such as Linear Queries and Compositions, the gains become substantially larger on structurally more complex queries. In particular, PARoG achieves significant improvements of 12.7% on Comparatives, 14.4% on Conjunctions, and 20.5 % on Superlatives. These results highlight that PARoG is especially effective in handling queries with multi-step reasoning and complex logical structures.

## 4.2 GENERALIZATION STUDY

To analyze the robustness across different schema organizations, we evaluate PARoG on CWQ and WebQSP with different source KGs (Freebase and WikiData), as shown in Table 3. Notably, the absolute performance on Wikidata is lower than that on Freebase because the datasets are originally annotated for Freebase. Moreover, Wikidata is substantially larger and more heterogeneous, which increases the difficulty for KG exploration and relation filtering. PARoG consistently produces substantial improvements over compared approaches under the WikiData setting. The results demonstrate that the proposed SPARQL-Guided Structured Planning and Plan-Answer-Refine are not tied to specific relations but generalize well under different schema organizations, relation granularities, and naming conventions. The improvements are especially pronounced on CWQ, where the queries are relatively more complicated.

Table 3: Generalization Study: performance of PARoG using different source KGs on WebQSP and CWQ.

| Method | WebQSP | CWQ |
|---|---|---|
| *with Freebase* | | |
| ToG | 76.2 | 57.1 |
| PoG | 82.0 | 63.2 |
| **PARoG** | **89.3** | **73.3** |
| *with WikiData* | | |
| ToG | 68.6 | 54.9 |
| PoG | 73.8 | 60.7 |
| **PARoG** | **79.1** | **69.5** |

### 4.3 ABLATION STUDY

We conduct ablation studies to examine the contributions of the two core components in our framework SPARQL-Guided Structured Planning and the Plan-Answer-Refine paradigm. The results are shown in Table 5 and 4. First, remove self-refinement consistently reduces performance across all datasets and settings. It can be observed the decline in performance when using GPT-3.5. This result demonstrates the Answer–Refine paradigm effectively mitigates error amplification, especially particularly in scenarios where the underlying LLM is relatively weak. Second, compare our SPARQL-supervised planning module (trained with Llama-3.1-8B) using much larger LLMs directly as planners. Despite having few parameters (8B), our model consistently outperforms GPT-3.5 (∼20B) and DeepSeek-R1 (671B) by large margins (up to 8.1 points on complex CWQ). This result demonstrates that SPARQL-guided supervision provides strong compositional reasoning signals, enabling smaller models to surpass much larger LLMs on reasoning path planning. These ablations prove that both self-refinement and SPARQL-supervised planning are essential to the effectiveness and efficiency of our framework.

To better understand the behavioral contribution of the Plan-Answer-Refine component, we further examine its ability to correct initially incorrect predictions. Specifically, for each dataset, we isolate all examples where the tentative answer produced by the vanilla LLM is incorrect, and compute the proportion of these errors that are successfully fixed after refinement. As shown in Table 6, PARoG corrects around 70% of the initial wrong answers and the correction rate is particularly high on GrailQA, which is consistent with its greater logical compositionality and schema diversity. This result highlights the effectiveness of the answer-refinement paradigm. The correctness rate of the initial answers is listed in Appendix G.

Table 4: Ablation Study: w/ or w/o Self-Refinement (*SR*).

| Method | WebQSP | GrailQA | CWQ |
|---|---|---|---|
| *GPT-3.5* | | | |
| w/o *SR* | 88.0 | 78.9 | 69.2 |
| w/ *SR* | **89.3** | **82.7** | **73.3** |
| *GPT-4* | | | |
| w/o *SR* | 89.7 | 85.5 | 77.2 |
| w/ *SR* | **91.2** | **87.2** | **79.3** |

### 4.4 EFFICIENCY ANALYSIS

We further analyze the efficiency of different methods in terms of LLM calls and token usage, as shown in Table 7. Our method consistently achieves not only higher accuracy but also greater efficiency across all datasets. These results demonstrate that PARoG not only advances the SOTA performance but also significantly reduces computational overhead, making it more efficient and cost-effective for real-world application.

Table 5: Ablation Study: Comparison our SPARQL-supervised planing module to LLMs.

| Method | # Para | WebQSP | GrailQA | CWQ |
|---|---|---|---|---|
| **Ours** | 8B | **89.3** | **82.7** | **73.3** |
| GPT-3.5 | ∼ 20B | 83.2 | 76.9 | 65.2 |
| Deepseek-R1 | 671B | 88.5 | 80.2 | 68.7 |

### 4.5 CASE STUDY

We also provide case studies to discuss the effectiveness and limitations of PARoG in Appendix J.

## 5 RELATED WORK

**LLMs with Knowledge Graphs.** Large Language Models (LLMs) have shown remarkable reasoning capabilities (Brown et al., 2020; Wei et al., 2022; Zhou et al., 2023) but often prone to hallucinate when answering knowledge-intensive queries (Ji et al., 2023). To address this, combining LLMs reasoning with external knowledge graphs (KGs) have been introduced (Logan et al., 2019; Luo et al., 2023; Jiang et al., 2023b; Pan et al., 2024). Approaches such as KG-based representation

Table 6: Error correction rate (CR Rate) of the Plan-Answer-Refine paradigm.

| Dataset | WebQSP | CWQ | GrailQA |
|---|---|---|---|
| **CR Rate** | 73.4 | 62.4 | 77.1 |

Table 7: Efficiency Analysis: Performance vs. token cost across different methods and datasets.

| Dataset | Method | LLM Call | Input Token | Output Token | Total Token | Hits@1 |
|---------|--------|----------|-------------|--------------|-------------|--------|
| **WebQSP** | ToG | 15.9 | 6,031.2 | 987.7 | 7,018.9 | 76.2 |
| | PoG | 9.0 | 5,234.8 | 282.9 | 5,517.7 | 82.0 |
| | **PARoG** | **8.3** | **5,012.3** | **241.4** | **5,253.8** | **89.3** |
| **CWQ** | ToG | 22.6 | 8,182.9 | 1,486.4 | 9,669.4 | 57.1 |
| | PoG | 13.3 | 7,803.0 | 353.2 | 8,156.2 | 63.2 |
| | **PARoG** | **10.2** | **7,110.7** | **288.5** | **7,398.8** | **73.3** |
| **GrailQA** | ToG | 11.1 | 4,066.0 | 774.6 | 4,840.6 | 68.7 |
| | PoG | 6.5 | 3,372.8 | 202.8 | 3,575.6 | 76.5 |
| | **PARoG** | **6.0** | **3,180.9** | **178.1** | **3,358.2** | **82.7** |

learning (Guu et al., 2020; Li et al., 2023b; Dehghan et al., 2024), knowledge-based instruction-tuning (Zhang et al., 2023; Chen et al., 2024a; Luo et al., 2024), retrieval-augmented generation with KG facts (Wang et al., 2024; Wen et al., 2024; Zhang et al., 2024; Wang et al., 2023b), graph-constrained generation (Guan et al., 2024; Luo et al., 2025), and semantic parsing on KGs (Ye et al., 2022; Yu et al., 2022) demonstrate the benefit of grounding LLM outputs in structured knowledge.

**Interactive LLM Reasoning over Knowledge Graphs.** Inspired by strong capability of deep reasoning on structured data (Jiang et al., 2023a; Edge et al., 2024; Jin et al., 2024), recent methods introduce explicit reasoning paths to guide LLM interactively inference over KGs and have achieved significant improvements (Yao et al., 2023; Li et al., 2024a; Mavromatis & Karypis, 2024; Sun et al., 2024; Tan et al., 2025; Chen et al., 2024b). Think-on-Graph (Sun et al., 2024) treats reasoning as agent-based exploration where LLMs iteratively search paths with traceability and correction. Generate-on-Graph (Xu et al., 2024) extends to incomplete KGs by enabling LLMs to generate missing triples. Plan-on-graph (Chen et al., 2024b) applies adaptive planning by decomposing questions into sub-goals and refining paths via guidance and reflection. KG-Agent (Jiang et al., 2025) formalizes multi-hop reasoning as program execution with tool use, KG execution, and memory updates. Debate-on-Graph (Ma et al., 2025) models reasoning as a multi-agent debate, where agents generate, and critique reasoning paths to enhance reliability. ReKnoS (Wang et al., 2025) introduces super-relations to connect relational paths, enabling bidirectional reasoning and improving retrieval efficiency. There are also more recent efforts such as (Shen et al., 2025) and (Zhu et al., 2025) introducing alignment and reflection-based strategies to regulate LLM reasoning over KGs.

Our work also belongs to this line of work but differs from prior methods by introducing SPARQL-guided structured planning and answer-refine mechanism. Compared with existing work, the proposed method enables reasoning over complex logical operations beyond linear paths, and explicitly mitigate the inconsistency between the parametric knowledge of LLM and external KG evidence.

## 6 DISCUSSION AND CONCLUSION

In this paper, we present Plan-Answer-Refine-on-Graph (PARoG) a novel framework for LLM reasoning over knowledge graphs. PARoG introduces two innovations SPARQL-guided structured planning and the Answer-Refinement paradigm, effectively mitigating search space truncation bias and error amplification issues. Extensive experiments on WebQSP, CWQ, and GrailQA demonstrate that PARoG achieves new state-of-the-art results while also being more efficient and cost-effective.

**Limitation.** Despite the improvements, PARoG still relies on the coverage and correctness of available KGs. Besides, while SPARQL-guided training reduces dependence on large models, generating high-quality planning data still requires strong teacher models (e.g., GPT-4o), which may limit accessibility. Moreover, our refinement process is static and offline without dynamic feedback loops during reasoning, we leave the exploration of online refinement for future work due to its complexity.

**Broader Impact and Future Work.** PARoG that structured symbolic guidance can enhance LLM reasoning, which can be applied wherever external structured signals are available beyond KGQA. In the future the study direct include dynamic refinement, multi-modal knowledge graphs, and bootstrapped self-improvement, which could make PARoG more scalable, general and accessible.

ACKNOWLEDGMENTS

This research/project is supported by the National Natural Science Foundation of China (No. 62302437) and State Street Zhejiang University Technology Center. We would also like to thank reviewers for their valuable comments.

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

## A    THE USE OF LARGE LANGUAGE MODELS (LLMS)

We used OpenAI ChatGPT for writings refinement and correction of typos during the preparation of this manuscript.

## B    ERROR ANALYSIS OF EXISTING METHODS.

We conduct an error analysis of existing methods (ToG and PoG). Among the failed examples:83.39% of WebQSP questions,85.34% of GrailQA questions, and 90.14% of CWQ queries fail due to missing answer entities in the retrieval phase. Figure 5 shows the distribution of different query types of the failed examples.

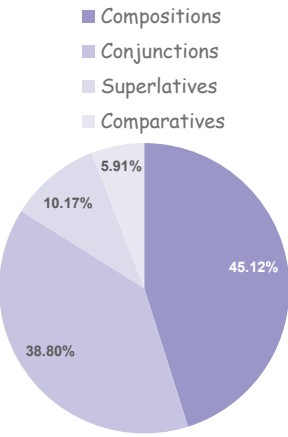

Figure 5: Error Distribution of PoG.

## C    PROMPT TEMPLATES FOR MODEL TRAINING

We describe the training input template used in our framework as follows:

```
Instruction:  Decompose the following complex question into a logical
sequence of simpler sub-questions.
Input:  Question:  [The original complex question]
Response:  1.  [First sub-question] 2.  [Second sub-question] 3.  ···
```

## D    ALGORITHM FOR THE PLAN-ANSWER-REFINE PARADIGM

We summarize the comprehensive algorithmic procedure of Plan-Answer-Refine as shown in Algorithm 1.

## E    DETAILS OF DATASETS

WebQSP consists of 4,737 natural language questions that require single or multi-hop reasoning over Freebase. GrailQA presents a more challenging scenario with 64,331 questions designed to test compositional generalization capabilities. CWQ contains 34,689 complex questions requiring multi-hop reasoning and constraint handling. These datasets collectively provide a robust testbed for evaluating various aspects of KGQA performance, including reasoning complexity, generalization ability, and scalability. The statistics of the datasets are summarized in Table 8.

---

**Algorithm 1** PARoG.

---

**Require:** Question $Q$, Knowledge Graph $\mathcal{G}$, LLM $\mathcal{M}$, Planning module $\mathrm{PLAN}(\cdot)$, instruction templates $I_A, I_R$, initial topic entity set $\mathcal{E}_0$ (size $n_0$), max iterations $T_{\max}$
**Ensure:** Final answer $a$
1: $\mathcal{O} \leftarrow \mathrm{PLAN}(Q)$         ▷ Generate sub-objectives with the SPARQL-supervised planner
2: $\mathcal{A} \leftarrow \emptyset$
3: **for each** sub-objective $o_i \in \mathcal{O}$ **do**
4:      $\hat{a}_i \leftarrow \mathcal{M}(Q, o_i, I_A)$           ▷ Tentative answer by parametric reasoning
5:      $\mathcal{P}_0 \leftarrow \{[\,]\}$            ▷ Initialize reasoning paths
6:      $\mathcal{E}_0 \leftarrow \{e_1^0, \ldots, e_{n_0}^0\}$
7:      **for** $t = 1$ **to** $T_{\max}$ **do**
8:         $\mathcal{E}_{t-1} \leftarrow \mathrm{TAILENTITIES}(\mathcal{P}_{t-1})$
9:         $\mathcal{R}_t^{init} \leftarrow \mathrm{NEIGHBORRELATIONS}(\mathcal{G}, \mathcal{E}_{t-1})$
10:        $\mathcal{R}_t \leftarrow \mathcal{M}(Q, o_i, \mathcal{O}, \mathcal{R}_t^{init})$     ▷ Filter relations with global plan awareness
11:        $\mathcal{E}_t^{cand} \leftarrow \mathrm{SPARQLQUERY}(\mathcal{G}, \mathcal{E}_{t-1}, \mathcal{R}_t)$
12:        $\mathrm{SCORE}(e) \leftarrow \mathcal{M}(Q, o_i, e), \ \forall e \in \mathcal{E}_t^{cand}$
13:        $\mathcal{E}_t \leftarrow \mathrm{SELECTRELEVANT}(\mathcal{E}_t^{cand}, \mathrm{SCORE})$   ▷ Select a variable number of most relevant entities
14:        $\mathcal{P}_t \leftarrow \mathrm{EXTENDPATHS}(\mathcal{P}_{t-1}, \mathcal{E}_t, \mathcal{R}_t)$
15:        $a_i \leftarrow \mathcal{M}(\mathcal{P}_t, o_i, \hat{a}_i, I_R)$           ▷ Self-refine tentative answer
16:        **if** $\mathrm{ALIGN}(\mathcal{P}_t, Q)$ = **false then**
17:           $a_i \leftarrow \hat{a}_i$            ▷ Fallback if retrieved knowledge is irrelevant
18:        **end if**
19:        **if** $\mathrm{SUFFICIENT}(a_i, Q)$ = **true then**
20:           **break**
21:        **end if**
22:      **end for**
23:      $\mathcal{A} \leftarrow \mathcal{A} \cup \{a_i\}$
24: **end for**
25: $a \leftarrow \mathrm{AGGREGATE}(\mathcal{A}, \mathcal{O})$     ▷ Combine refined sub-answers according to $\mathcal{O}$ to form the final answer.
26: **return** $a$

---

Table 8: Statistics of datasets.

| Datasets | #Train | #Test | Max #hop |
|---|---|---|---|
| WebQSP | 2,826 | 1,628 | 2 |
| CWQ | 27,639 | 3,531 | 4 |
| GrailQA | 44,337 | 13,231 | 4 |

## F    DETAILS OF COMPARED BASELINES

We compare PARoG with 17 LLM-based approaches from 3 categories: (1) LLM prompting methods, (2) LLM reasoning over KGs (LLM $\otimes$ KG), and (3) end-to-end fine-tuned KG-augmented LLMs. The details of the compared approaches are described as follows.

### F.1    LLM PROMPTING

- **Input-Output Prompting** (Brown et al., 2020):A standard few-shot prompting approach without explicit reasoning guidance, serving as a basic LLM QA baseline.

- **Chain-of-Thought** (Wei et al., 2022): Chain-of-Thought prompting encourages the LLM to explicitly generate intermediate reasoning steps, improving logical consistency on complex queries.

- **Self-Consistency** (Wang et al., 2023c): Self-consistency prompting samples multiple reasoning chains and aggregates their results, reducing random errors and improving answer stability.

## F.2 LLM ⊗ KGs

- **Think-on-Graph** (Sun et al., 2024): Think-on-Graph models reasoning as an agent-based exploration, where the LLM iteratively traverses the knowledge graph to build interpretable paths.

- **Reasoning-on-Graph** (Luo et al., 2024): Reason-on-Graph constrains LLM outputs to faithful graph-grounded reasoning paths, improving interpretability and correctness of answers.

- **KG-Agent** (Jiang et al., 2025): An autonomous agent framework that formalizes multi-hop reasoning as program execution with KG queries, external tool use, and memory updates.

- **StructGPT** (Jiang et al., 2023a): A structured generation framework where LLMs are guided by schema-constrained prompts to produce reasoning paths over structured data.

- **Plan-on-Graph** (Chen et al., 2024b): Plan-on-Graph decomposes complex queries into structured sub-goals and adaptively plans reasoning paths on the KG, enabling better compositional reasoning.

- **ReKnowS** (Wang et al., 2025): ReKnowS introduces the concept of super-relations to connect multiple relational paths, allowing bidirectional reasoning and improving retrieval efficiency.

- **KB-BINDER** (Li et al., 2023a): KB-BINDER bridges LLM reasoning with KG facts using a binding mechanism that grounds parametric knowledge in structured evidence.

- **Debate-on-Graph** (Ma et al., 2025): Debate-on-Graph models reasoning as a multi-agent debate, where different agents generate and critique reasoning paths to improve reliability.

## F.3 END-TO-END FINE-TUNED KG-AUGMENTED LLMS

- **RnG-KBQA** (Ye et al., 2022): A generation-augmented KBQA model that iteratively ranks candidate answers, combining generative reasoning with retrieval.

- **TIARA** (Shu et al., 2022): A multi-grained retrieval framework designed to strengthen robustness of KBQA systems against noisy or incomplete evidence.

- **FC-KBQA** (Zhang et al., 2023): Fine-to-Coarse composition framework that first retrieves broad candidates and then refines answers hierarchically for complex KBQA.

- **Pangu** (Gu et al., 2023): An end-to-end KBQA model that emphasizes compositional generalization, allowing it to handle more complex query structures.

- **FlexKBQA** (Li et al., 2024b): A flexible, LLM-powered KBQA framework designed for few-shot learning and adaptable to low-resource settings.

- **GAIN** (Shu & Yu, 2024): A KBQA method optimized for distribution shifts, making reasoning more robust across different domains and data splits.

## F.4 GRAPH-RETRIEVAL METHODS

- **GNN-RAG** (Mavromatis & Karypis, 2025): A deep learning method uses graph neural networks to retrieve the most relevant nodes and subgraphs for LLM reasoning.

- **SubgraphRag** (Li et al., 2025): A method builds and retrieves localized subgraphs to provide structured graph context for LLMs.

## G STATISTICAL ANALYSIS OF MAJOR COMPONENTS

We conduct analysis to measure the proportion of questions for which the search process prematurely prunes the correct path (reported as truncation rate). The results are as follows.

Table 9: Truncation Rate.

| Method | Truncation Rate (%) |
|--------|---------------------|
| PoG    | 38.82               |
| PARoG  | 19.48               |

Table 10: Correctness rate of initial answers (%).

| Dataset | WebQSP | CWQ | GrailQA |
|---|---|---|---|
| **Correctness rate of initial answers (%)** | 61.7 | 28.9 | 43.4 |

Moreover, to better understand how often the tentative answers produced by the LLM are incorrect, we list the correctness rate (%) in the initial answers generated by LLM as shown in Table 10.

These numbers verify that LLM parametric knowledge is often incomplete or unreliable, highlighting the necessity of a refinement stage grounded in KG execution.

## H    STABILITY AND SENSITIVITY ANALYSIS

To analyze the stability of the proposed method, we run the experiments under three seeds and report the mean $\pm$ 95% CI for different settings. The results are shown in Table 11, 12 and 13.

Table 11: Stability Analysis on main datasets.

| Method | WebQSP | GrailQA | CWQ |
|---|---|---|---|
| *GPT-3.5* | | | |
| ToG | $76.4 \pm 1.9$ | $68.9 \pm 1.5$ | $57.2 \pm 1.8$ |
| PoG | $82.1 \pm 2.2$ | $76.5 \pm 2.1$ | $63.2 \pm 1.2$ |
| **PARoG** | $\mathbf{89.0} \pm 1.3$ | $\mathbf{82.7} \pm 1.5$ | $\mathbf{73.1} \pm 0.9$ |
| *GPT-4* | | | |
| ToG | $80.7 \pm 1.7$ | $80.9 \pm 1.4$ | $65.6 \pm 2.2$ |
| PoG | $87.3 \pm 1.5$ | $84.3 \pm 1.8$ | $75.0 \pm 1.4$ |
| **PARoG** | $\mathbf{91.2} \pm 0.9$ | $\mathbf{87.1} \pm 1.3$ | $\mathbf{79.3} \pm 1.1$ |

Table 12: Stability analysis on w/ or w/o Self-Refinement (SR).

| Method | WebQSP | GrailQA | CWQ |
|---|---|---|---|
| *GPT-3.5* | | | |
| w/o *SR* | $88.0 \pm 1.6$ | $78.9 \pm 1.4$ | $69.2 \pm 1.2$ |
| w/ *SR* | $\mathbf{89.0} \pm 1.3$ | $\mathbf{82.7} \pm 1.5$ | $\mathbf{73.1} \pm 0.9$ |
| *GPT-4* | | | |
| w/o *SR* | $89.7 \pm 1.1$ | $85.5 \pm 1.4$ | $77.2 \pm 1.0$ |
| w/ *SR* | $\mathbf{91.2} \pm 0.9$ | $\mathbf{87.1} \pm 1.3$ | $\mathbf{79.3} \pm 1.1$ |

Notably, PARoG shows higher accuracy and consistently smaller confidence intervals than all baselines. Moreover, the smaller planner outperforms larger general LLMs. Even the smallest 1B planner still achieves comparable performance compared to Deepseek-R1, confirming that SPARQL-guided training provides a stronger planning signal than scaling model size alone. These results prove the stability and reproducibility of the SPARQL-guided planner and the Answer-Refine mechanism.

## I    EFFICIENCY TRADE-OFFS.

To better characterize the efficiency–accuracy trade-off, we conducted an additional study varying the maximum planning iterations in PARoG as shown in Figure 6. Unlike existing methods, PARoG does not use a fixed beam width. Instead, it maintains a dynamic beam, while the overall search budget is controlled by the maximum number of planning iterations.

Table 13: Stability and Sensitivity analysis on different planners.

| Method | # Para | WebQSP | GrailQA | CWQ |
|---|---|---|---|---|
| *General LLMs* | | | | |
| GPT-3.5 | $\sim$ 20B | 83.1 ± 1.4 | 76.7 ± 1.6 | 65.4 ± 1.2 |
| Deepseek-R1 | 671B | 88.2 ± 1.5 | 80.2 ± 1.3 | 68.7 ± 1.1 |
| *Ours* | | | | |
| **llama3.1-8B** | 8B | **89.0** ± 1.3 | **82.7** ± 1.5 | **73.3** ± 0.9 |
| **llama3.2-1B** | 1B | 87.2 ± 1.2 | 79.1 ± 1.6 | 68.5 ± 1.1 |
| **llama2-13B** | 13B | 88.4 ± 1.4 | 81.9 ± 1.2 | 70.3 ± 1.0 |

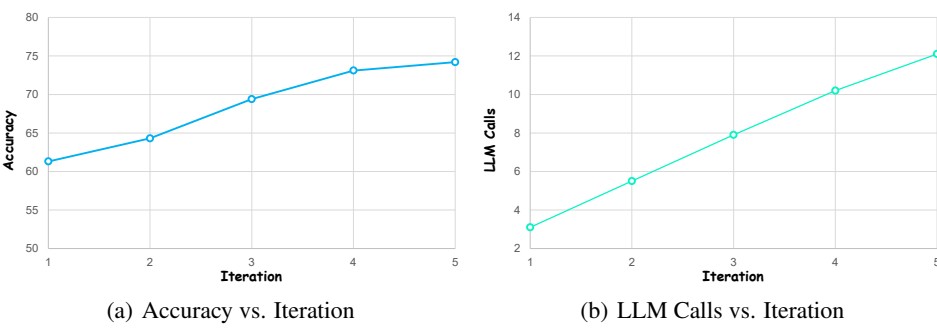

(a) Accuracy vs. Iteration  (b) LLM Calls vs. Iteration

Figure 6: Efficiency Trade-offs.

## J CASE STUDY

To better understand the reasoning improvements brought by our proposed method, we analyze two representative failure cases from the baseline method Planning-on-Graph (PoG) and contrast them with the corresponding inference paths of our model (PARoG), as shown in Table 14.

### J.1 CASE 1: COMPOSITION PLAN ERROR

This example investigates the question:

> What is considered modern in the country where Bilady, Bilady, Bilady language is the national anthem?

The question requires a two-hop reasoning process: (1) identify the country given the national anthem, and (2) determine the language that is considered "modern" in that country. The baseline PoG fails in the planning stage by misinterpreting the intent of "modern" as referring to the country itself rather than a language. As a result, it outputs "Egypt" as the final answer, which is incomplete and incorrect.

In contrast, PARoG correctly preserves the linguistic intent of the original question during the planning phase. Unlike PoG, which misinterprets "modern" as referring to a modern country, PARoG correctly interprets it as referring to a modern language spoken in the identified country.

It first identifies Egypt as the country where "Bilady, Bilady, Bilady" is the national anthem—same as PoG—but goes a step further by reasoning that Modern Standard Arabic is the relevant modern language spoken in Egypt. This is captured in the following reasoning path:

Egypt → languages spoken → Modern Standard Arabic

By grounding the abstract query term "modern" into a specific linguistic attribute, PARoG successfully answers the question with Modern Standard Arabic, demonstrating its ability to disambiguate vague terms and construct semantically aligned plans that lead to correct, complete answers.

This case highlights the strength of the Plan-Answer-Refine framework in maintaining semantic consistency and avoiding reasoning drift during multi-hop KGQA.

Table 14: Examples of reasoning conducted by PARoG. Red denotes incorrect reasoning paths and answers, while green denotes correct ones.

| **Case 1**: Composition plan error | |
|---|---|
| Question | What is considered modern in the country where Bilady, Bilady, Bilady language is the national anthem? |
| Answer | Modern Standard Arabic |
| PoG | # Plan: [Identify the country where "Bilady, Bilady, Bilady" is the national anthem, Research the definition of "modern" in the context of countries]
# Reasoning Path: m.0478lbx → government.national_anthem_of_a_country.anthem → Bilady, Bilady, Bilady
m.0478lbx → government.national_anthem_of_a_country.country → Egypt
# Answer: Egypt |
| PARoG | # Plan: [Identify the country where "Bilady, Bilady, Bilady" is the national anthem, Research modern cultural, social, or technological aspects in that country]
# Reasoning Path: m.0478lbx → government.national_anthem_of_a_country.anthem → Bilady, Bilady, Bilady
m.0478lbx → government.national_anthem_of_a_country.country → Egypt
Egypt → location.country.languages_spoken → Modern Standard Arabic
# Answer: Modern Standard Arabic |
| **Case 2**: Parametric Knowledge Bridging KG Gaps | |
| Question | What movies did Adam Sandler play in and is about Christmas? |
| Answer | Eight Crazy Nights |
| PoG | # Reasoning Path: Adam Sandler → film.actor.film → {Eight Crazy Nights,The Chanukah Song, Reign Over Me, Funny People, The Meyerowitz Stories, The Week Of}
# Answer: The Chanukah Song |
| PARoG | # Reasoning Path: Adam Sandler → film.actor.film → {Eight Crazy Nights, Funny People, Reign Over Me, etc.}
LLM answers:Eight Crazy Nights is about Christmas
# Answer: Eight Crazy Nights |

## J.2 CASE 2: PARAMETRIC KNOWLEDGE BRIDGING KG GAPS

We also examine the following question:

> What movies did Adam Sandler play in and is about Christmas?

The PoG baseline retrieves several films involving Adam Sandler—such as The Chanukah Song, Reign Over Me, and Funny People—but it fails to search whether these movies are related to Christmas, ultimately yielding the incorrect result The Chanukah Song. In contrast, PARoG follows a similar retrieval process and also identifies several films involving Adam Sandler through its agent module. However, despite failing to retrieve any explicit evidence from the knowledge graph regarding the "Christmas" constraint, PARoG demonstrates a more robust and semantically grounded refinement process. By leveraging the parametric knowledge encoded within the LLM, it successfully infers that Eight Crazy Nights is a Christmas-themed film. This case illustrates the strength of the plan-answer-refine paradigm in producing correct answers even when symbolic evidence from the KG is absent, showcasing the complementary power of LLM-based reasoning.

## K FAILURE CASE ANALYSIS

Table 15 presents representative failure cases of PARoG and highlights the underlying causes across three major error categories. Case 1 illustrates a relation-selection error, where the planner or exploration stage selects a relation chain which is semantically plausible but ultimately incorrect(e.g.,

Table 15: Failure cases of PARoG and corresponding error causes.

| **Case 1**: Relation selection error | |
|---|---|
| Question | What movie did Ron Howard do with cinematography was by Mark Irwin? |
| Ground truth | Osmosis Jones |
| error cause | # PARoG Reasoning Path: Mark Irwin → film.cinematographer.film →Scream
Ron Howard" → film.film.directed_by→Apollo 13
Answer:Scream
# Ground truth path: Ron Howard → film.actor.film → m.03g24
m.03g24 → film.performance.film → Osmosis Jones
Osmosis Jones → film.film.cinematography → Mark Irwin
# Cause: Incorrect relation selection (choosing film.film.directed_by and film.cinematographer.film instead of the actor–performance relations) diverts exploration to irrelevant paths, thereby missing the gold answer. |
| **Case 2**: Incomplete multi-answer retrieval (partial answer set) | |
| Question | who plays blaine in batman |
| Ground truth | Matthew Wagner,Danny Trejo,Carlos Alazraqui,Tom Hardy |
| error cause | # PARoG reasoning path (only one branch retrieved):
# Film path retrieved:
Blaine → film.film_character.portrayed_in_films →
film.performance.actor → Tom Hardy, Carlos Alazraqui, Matthew Wagner
Missing actor from the TV branch: Danny Trejo.

# Ground truth path (union of two branches):
# Film path:
Blaine → film.film_character.portrayed_in_films →
film.performance.actor → Tom Hardy, Carlos Alazraqui, Matthew Wagner
# TV path:
Blaine → tv.tv_character.appeared_in_tv_program →
tv.regular_tv_appearance.actor → Danny Trejo

# Cause: The ground-truth answer set is formed by the union of a film path and a TV path, but PARoG identifies only one branch (film) and overlooks the tv_program branch, resulting in an incomplete multi-answer retrieval. |
| **Case 3**: Path absence due to KG incompleteness. | |
| Question | What countries in Oceania, that have the emissions per capita in dated metric ton of 0.091351? |
| Ground truth | Papua New Guinea |
| error cause | PARoG Reasoning: Cook Islands is a country in Oceania that has emissions per capita of 0.091351 metric tons
Path absence due to KG incompleteness. The KG does not contain a valid relational path that instantiates the numeric condition 0.091351 (i.e., this value is missing/not represented), so PARoG cannot retrieve the exact matching entity and is forced to Use the LLM internal Knowledge. |

directed_by instead of actor-performance relations), diverting the search away from the gold path. Case 2 shows an instance of incomplete multi-branch retrieval, where the system successfully recovers the film-related path but misses an additional TV-series branch required to produce the full multi-answer set. This demonstrates that although PARoG effectively handles compositional constraints, multi-source answer aggregation remains challenging when evidence spans heterogeneous subgraphs. Case 3 shows a failure case caused by KG incompleteness: when numerical attributes or relational paths needed to instantiate a constraint are absent from the underlying KG, PARoG is unable to retrieve the gold entity and must instead rely on parametric knowledge, often leading

to incorrect final predictions. Overall, these cases reveal that remaining errors primarily stem from (i) subtle relation disambiguation, (ii) multi-branch retrieval coverage, and (iii) missing or incomplete KG facts—providing actionable directions for future improvements such as stronger relation grounding, expanded query branching, and KG-aware confidence estimation.

# L    SEARCH SPARQL

we define several SPARQL queries for Freebase queries, which can be executed to search the relation and entity in the Knowledge graph

```
PREFIX ns: <http://rdf.freebase.com/ns/>
SELECT ?relation
WHERE {
    ns:mid ?relation ?x .
}
```

```
PREFIX ns: <http://rdf.freebase.com/ns/>
SELECT ?relation
WHERE {
    ?x ?relation ns:mid .
}
```

## L.1    ENTITY SEARCH

```
    PREFIX ns: <http://rdf.freebase.com/ns/>
    SELECT ?tailEntity
    WHERE {
        ns:mid ns:relation ?tailEntity .
    }
```

```
PREFIX ns: <http://rdf.freebase.com/ns/>
SELECT ?tailEntity
WHERE {
    ?tailEntity ns:mid ns:relation  .
}
```

## L.2    ENTITY NAME

```
PREFIX ns: <http://rdf.freebase.com/ns/>
SELECT DISTINCT ?tailEntity
WHERE {
    {
        ?entity ns:type.object.name ?tailEntity .
        FILTER(?entity = ns:mid)
    }
    UNION
    {
        ?entity <http://www.w3.org/2002/07/owlsameAs> ?tailEntity .
        FILTER(?entity = ns:mid)
    }
}
```

# M    PROMPT TEMPLATES FOR LLM AGENTS

We introduce the full prompting strategy used in our framework, which can be divided into two main stages:

- **Data Generation Stage:** Generating training data from SPARQL queries.
- **Agent Reasoning Stage:** Guiding the LLM through the full Plan-Answer-Refine reasoning procedure based on decomposed subgoals and retrieved knowledge.

## M.1   DATA GENERATION STAGE:

For each input question, GPT-4o produces a set of decomposed sub-questions that reflect the logical structure of the underlying SPARQL query.Here we display the prompt we use to generate the sub-question.

### M.1.1   DATA GENERATE

```
Please break down the process of answering the question into as few
    subobjectives as possible based on semantic analysis and sparql

Now you need to directly output subobjectives of the following question
    in list format like the example above. The output format should be [
    subobjective1, subobjective2,...]

Q: \{Query\}

Sparql: \{Sparql Query\}

Output:
```

## M.2   AGENT REASONING STAGE:

We detail the complete prompt templates used in our iterative reasoning framework, including answer initialization, relation/entity pruning, state updating, and self-refinement.

## M.3   INIT ANSWERING

```
ased on your own knowledge, output the current known information required
    to achieve the subobjectives.

\texttt{In-Context Few-shot}

Q: \{Query\}

Subobjectives:\{list of sub questions\}

Now you need to directly output the results of the following question in
    JSON format without other information or notes.

Output:
```

## M.4   RELATION PRUNE

```
Please provide as few highly relevant relations as possible to the
    question and its subobjectives from the following relations.

\texttt{In-Context Few-shot}

Q: \{Query\}

Subobjectives:\{list of sub questions\}

Topic Entity: \{Topic Entity\}
```

```
Relations: \{list of relations\}

output:
```

The LLM is instructed to directly select a subset of candidate relations(No thresholds), we do not set a fixed thresholds.and this discrete selection is used as the pruned relation set; we do not apply any additional numeric thresholds on LLM scores.

## M.5 ENTITY PRUNE

For each triple pattern (e,r,?)(e, r, ?)(e,r,?), we construct a prompt of the form.

```
Which entities in the following list ([] in Triples) can be used to
    answer question? Please provide the minimum possible number of
    entities, and strictly adhering to the constraints mentioned in the
    question.
Now you need to directly output the entities from [] in Triplets for the
    following question in list format without other information or notes.

\texttt{In-Context Few-shot}

Q: \{Query\}

Subobjectives:\{list of sub questions\}

Relation: \{Current Relation\}

Entites:  \{list of entities\}

output:
```

The LLM is prompted to directly choose a subset of the provided candidate entities that are most relevant for answering the question(no thresholds).similarly,we do not set a fixed threshold.

## M.6 UPDATE THE KG REASONING STATE

```
Based on the provided information to revise the memory,if the memory has
    conflict with the provided information,use the provided information
    to revise the memory.If the provided information is not enough to
    revise the memory, keep the memory unchanged.

\texttt{In-Context Few-shot}

Now you need to directly output the results of the following question in
    JSON format without other information or notes.

Q: \{Query\}

Memory: \{the status of the sub-questions\}

Knowledge triples: \{Explored Paths\}

output:

Q: \{Query\}

Knowledge triples: \{Explored Paths\}

Output:
```

### M.7 SELF-REFINE

```
Given the question and the associated retrieved knowledge graph triples (
    entity, relation, entity), you are asked to
self-refine the initial answers based on them. if the initial answers
    have conflict with the provided information,use the provided
    information to refine them.If the provided information is not enough
    to refine the answers, keep the answers unchanged.

\texttt{In-Context Few-shot}

Q: \{Query\}

Knowledge triples: \{Explored Paths\}

Inital answers:\{The inital answers generated by the llm\}

Output:
```

### M.8 ANSWER

PARoG runs an iterative plan–answer–refine loop with two complementary stopping conditions. First, we cap the planning horizon by a fixed maximum number of iterations. Second, at each iteration we call a final-answer head that takes the original question, the currently explored KG paths, and the refined result of resolved sub-questions as input.

```
Please answer the question based on the memory, related knowledge
    triplets and your knowledge.

Now you need to directly output the results of the following question in
    JSON format (must include "A" and "R") without other information or
    notes.

\texttt{In-Context Few-shot}

Q: \{Query\}

Knowledge triples: \{Explored Paths\}

Memory: \{the status of the sub-questions\}

Output:
```

The model is required to output a JSON object with keys "A" (final answer) and "R" (short rationale). If the predicted answer field "A" is non-empty and not an "unknown" placeholder, we accept it and terminate the whole process early, without further KG exploration. Otherwise, we continue to the next iteration until either a satisfactory answer is produced or the maximum iteration is reached, in which case we fall back to answering based on the LLM's own parametric knowledge plus the accumulated triples in the context.

