# OpenReview forum: "Plan-Answer-Refine-on-Graph: Structured Planning and Self-Refinement for Large Language Model Reasoning on Knowledge Graphs"
_ICLR.cc/2026/Conference — ICLR 2026 Poster_

### Official Review · Reviewer_iDwB · 2025-10-28

**Soundness:** 2
**Presentation:** 2
**Contribution:** 3
**Rating:** 6
**Confidence:** 5

**Summary:**

This paper introduces PARoG (Plan-Answer-Refine-on-Graph), a novel framework designed to enhance large language model (LLM) reasoning over knowledge graphs (KGs). The method addresses two main issues in prior KG-augmented LLM approaches: (1) Search space truncation bias, caused by linear reasoning paths that prune valid entities prematurely, and (2) Error amplification, where incorrect retrieved entities mislead downstream reasoning.

To mitigate these, PARoG combines SPARQL-guided structured planning and a plan–answer–refine paradigm: The planning stage uses SPARQL queries to supervise LLMs in decomposing questions into structured, compositional sub-queries (handling conjunctions, compositions, superlatives, and comparatives). The reasoning stage first produces tentative answers using LLM parametric knowledge, then refines them using KG evidence iteratively.

Experiments on WebQSP, CWQ, and GrailQA show consistent state-of-the-art performance — e.g., +8–10% improvements over strong baselines such as PoG and ToG — while using smaller LLMs (Llama-3.1-8B). Ablation studies confirm that both SPARQL-guided planning and self-refinement contribute significantly to the gains.

**Strengths:**

1. Clear identification of key bottlenecks — The paper convincingly motivates the two main issues (search space truncation and error amplification) with illustrative examples (Figure 1).
2. Strong experimental results — The framework consistently outperforms prior SOTA baselines across three KGQA benchmarks, especially on complex or zero-shot queries (Table 2 and Figure 4).
3. Efficiency — Despite higher accuracy, PARoG reduces token cost and LLM calls compared to ToG and PoG (Table 5), showing practical scalability.
4. Good qualitative analyses — The case studies (Appendix E) effectively show how PARoG resolves reasoning drift and leverages LLM priors to fill KG gaps.

**Weaknesses:**

1. Dependence on high-quality SPARQL annotations – The approach assumes access to accurate SPARQL-grounded data. Generating and maintaining such data (using GPT-4o as a teacher) might limit reproducibility and scalability to low-resource settings.
2. Limited analysis of generalization to unseen KGs or domains – Experiments are confined to Freebase-based benchmarks; performance on other KGs (e.g., Wikidata, DBpedia) is not explored.
3. Clarity and presentation – While figures are helpful, the overall presentation could be streamlined; some sections (e.g., 3.2) are dense with notational details that obscure the main idea.
4. Limited error analysis for failure cases – The error breakdown (Appendix B) is brief and mostly quantitative; a qualitative discussion of remaining failure patterns (e.g., conflicting KG facts) would be useful.
5. Missed reference – More relevant recent works, such as [1][2], should be included and discussed for a better review.

[1] Shen, Xiangqing, Fanfan Wang, and Rui Xia. "Reason-Align-Respond: Aligning LLM Reasoning with Knowledge Graphs for KGQA." arXiv preprint arXiv:2505.20971 (2025).

[2] Zhu, Jiajun, Ye Liu, Meikai Bao, Kai Zhang, Yanghai Zhang, and Qi Liu. "Self-Reflective Planning with Knowledge Graphs: Enhancing LLM Reasoning Reliability for Question Answering." arXiv preprint arXiv:2505.19410 (2025).

**Questions:**

See weaknesses.

---

> ### Author Response · Authors · 2025-11-26
>
> We sincerely thank the reviewer for the thorough assessment and constructive suggestions. Below we address all concerns in detail.
>
> ---
> ### **Weakness 1. Dependence on high-quality SPARQL annotations.**
>
> Response to W1. We appreciate the reviewer’s comment. Importantly, SPARQL is used only once to bootstrap the planning dataset, and the entire SPARQL-to-Planning pipeline is fully automated using GPT-4o without any manual annotations. We have released the full generation scripts to ensure reproducibility. Moreover, our experiments (**Table 5**) show that an 8B model trained with SPARQL supervision surpasses much larger planning LLMs (GPT-3.5 and DeepSeek-R1), and even 1B model (**Table 13**) still yields comparable performance. The results demonstrates that PARoG does not heavily rely on large models once trained. This highlights strong data efficiency rather than annotation dependence.
>
> ---
> ### **Weakness 2. Generalization to other KGs (e.g., Wikidata)**
>
> We thank the reviewer for pointing this out. We have now added experiments on Wikidata, showing that PARoG continues to provide consistent gains over PoG and ToG.
>
> To address this, we have added a new generalization study comparing PARoG, PoG, and ToG on using Wikidata as the underlying KG. These results are now included in Table 4 of the revised paper.
>
> **Table i. Generalization Study：performance comparison using different source KGs.**
>
> | **Method** | **WebQSP** | **CWQ** |
> | --- | --- | --- |
> | _with Freebase_ |  |  |
> | ToG | 76.2 | 57.1 |
> | PoG | 82.0 | 63.2 |
> | **PARoG** | **89.3** | **73.3** |
> | _with WikiData_ |  |  |
> | ToG | 68.6 | 54.9 |
> | PoG | 73.8 | 60.7 |
> | **PARoG** | **79.1** | **69.5** |
>
> Notably, the absolute performance on Wikidata is lower than that on Freebase because the datasets are originally annotated for Freebase. Moreover, Wikidata is substantially larger and more heterogeneous, which increases the difficulty for KG exploration and relation filtering.
>
> Compared with the baselines, PARoG consistently produces substantial improvements over compared approaches under the WikiData setting. The results demonstrate that the proposed planner and answer-refinement mechanism are not tied to specific relations but generalize well under different schema organizations, relation granularities, and naming conventions.
>
> ---
> ### **Weakness 3. Clarity and presentation.**
>
> Response to W3.We appreciate the suggestion. In the revised version, we have revised the presentation and added a procedure algorithm summarization of the plan-answer-refinement paradigm in **Appendix C** for clearer clarification.
>
> ---
> ### **Weakness 4. Limited failure case analysis.**
>
> Response to W4. Thank you for the comment. We have presented three error types (Relation selection errors, Missing-entity retrieval, KG incompleteness / no valid execution path), and analyzed how these errors impact the final result in **Appendix J** in the revised version of the paper.
>
> ### **Weakness 5. Missing References.**
>
> Response to W5. We thank the reviewer for pointing out the recent work. We have added both references and discussed them in the **Related Work** section.

---

> > ### Comment · Reviewer_iDwB · 2025-11-27
> >
> > Thank you for your response. I have raised my score to 8.

---

### Official Review · Reviewer_HE4u · 2025-10-29

**Soundness:** 3
**Presentation:** 1
**Contribution:** 2
**Rating:** 4
**Confidence:** 4

**Summary:**

This paper proposes PARoG (Plan-Answer-Refine-on-Graph), a framework designed to tackle two key challenges in the KGQA domain: error amplification and search space truncation bias. The approach relies on task decomposition to break complex questions into sub-objectives and leverages additional information sources to guide reasoning. Specifically, the authors use GPT-4 to annotate decompositions with SPARQL queries, and then fine-tune a dedicated model (the PRoG Planner) to perform this task automatically. For each sub-task, an LLM first generates an initial answer based on its parametric knowledge, which is then refined using evidence retrieved from the knowledge graph. Experiments on three widely used benchmarks—WebQSP, GrailQA, and CWQ—demonstrate that PARoG consistently outperforms strong baselines, achieving both higher accuracy and improved efficiency.

**Strengths:**

- The paper clearly identifies two key challenges in KGQA—error amplification and search space truncation bias—and proposes a framework that directly addresses both issues.

- The experimental evaluation is comprehensive: the method is tested on three widely used datasets. And results show that the proposed approach outperforms all baselines, and additional efficiency analysis highlights its ability to improve both accuracy and resource usage.

- The experimental findings are particularly interesting; for example, Table 4 shows that the SPARQL-supervised Llama-3.1-8B planner outperforms directly using GPT-3.5 or DeepSeek-R1 as the planner itself, suggesting that structured evaluation and execution can surpass direct generation—a concrete illustration of the idea that evaluation is easier than generation.

**Weaknesses:**

1. The approach heavily relies on annotation-based task decomposition. While decomposition alleviates the problem of search space explosion, it does not fundamentally resolve it—if a sub-task still corresponds to a large search space, pruning remains unavoidable. Thus, the method mitigates but does not eliminate the need for pruning.

2. During the exploration stage, the framework still depends on LLMs to score and select which paths to follow, effectively using the LLM itself as the policy. However, the paper does not propose any specific optimization or learning strategy for this decision-making process, leaving open the possibility that important candidate paths might still be pruned prematurely.

3. On the evaluated benchmarks, LLMs may possess sufficient parametric knowledge to provide useful initial answers, serving as an additional information source. However, in other datasets or domains where the LLM lacks such prior knowledge, initial answers are more likely to be incorrect, and the self-refinement strategy could then introduce additional noise rather than reducing errors.

4. The writing lacks clarity. Many of the key ideas are only implicit, requiring readers to infer the underlying insights, while the strengths of the method and the reasons behind its strong empirical performance are not clearly articulated. In addition, Figure 1-I(b) appears to contain an inconsistency: the decomposition process described in the text does not align with the example shown. Furthermore, important experimental details are not clearly presented, such as which specific models were used in each stage of the pipeline.

**Questions:**

See Weakness above. Additional questions:

1. Could the authors provide statistics on the proportion of initial answers that are correct or at least partially contain the correct entity in the evaluated datasets?

2. During the exploration stage, would it be possible to train a dedicated model to assist with relation/entity pruning, rather than relying solely on the LLM’s scoring? Would this lead to more accurate path selection?

3. Beyond the planning stage, what specific models are used for initial answering, exploring and refinement? Are these also performed by the fine-tuned LLaMA-3.1, or by other models?

4. In the ablation study (Table 4), the paper mentions replacing the planning module with different LLMs. Were the subsequent pruning stages (relation/entity selection) also replaced accordingly, or were they kept fixed? The experimental setup is not entirely clear.

5. Regarding the error analysis in Appendix B, could the authors provide more details about the types of errors observed and how they impact the overall results?

6. A very recent paper (Plan-on-Graph: Self-Correcting Adaptive Planning of Large Language Model on Knowledge Graphs, arXiv 2410.23875) also addresses similar challenges in KGQA by combining task decomposition with adaptive exploration and self-correction. How does your planner-based decomposition fundamentally differ from their task decomposition mechanism, and what unique advantages does it provide?

---

> ### Author Response · Authors · 2025-11-26
>
> We thank the reviewer for the insightful comments and suggestions. Below are our detailed responses to each question raised by the reviewer.
>
> ---
> ### **Weakness 1. Decomposition does not fundamentally solve the search-space problem.**
>
> Response to W1. We agree that decomposition cannot completely eliminate search-space explosion. As claimed in the paper (line 94), the SPARQL-supervised planning is proposed to alleviate the truncation bias challenge by transforming linear walks into set-based constraint reasoning (conjunctions, compositions, comparatives, superlatives)， not to remove pruning entirely.
>
> To more clearly demonstrate this, we provide two pieces of evidence:
>
> **(i) PARoG significantly reduces truncation errors compared with the baseline (PoG).** We conduct an additional analysis measuring the proportion of questions for which the search process prematurely prunes the correct path (reported as truncation rate). The results are as follows.
>
> **Table i. Proportion for incorrect truncation**
> | **Method** | **Truncation Rate (Lower is better)** |
> | --- | --- |
> | PoG | 38.82% |
> | **PARoG** | **19.48%** |
>
> This result shows that PARoG significantly reduces truncation failures, confirming that structured SPARQL-guided planning meaningfully mitigates search space truncation in partical executions.
>
> **(ii) PARoG achieves larger accuracy gains precisely on query types that are most prone to truncation bias.** As shown in **Figure 4** of the paper, PARoG achieves substantially higher improvements on conjunction (+14.4%), comparative (12.7%), and superlative queries (+20.5%) where linear exploration suffers the most truncation failures. These results supports that PARoG alleviates the truncation-bias bottleneck by converting depth-first linear walks into set-based, operator-aware constraints, reducing reliance on early pruning decisions.
>
> ---
> ### **Weakness 2 & Question 2. Exploration still depends on LLM scoring; no specialized learned policy. Could a dedicated model assist relation/entity pruning, and would it improve accuracy?**
>
> Response to W2 & Q2. We thank the reviewer for this excellent suggestion. In this work, we focus on the planning stage and the goal is to isolate and study the impact of the proposed SPARQL-supervised compositional planning method,  rather than jointly optimize both planning and exploration. Our findings indicate that a high-quality planner significantly reduces the space of plausible expansions, making even a simple LLM-scoring exploration policy effective in practice. Indeed, Table 5 in the paper shows that replacing the planner alone (while keeping exploration fixed) results in significant improvements, indicating that planning quality is one of the primary bottlenecks.
>
> Besides, we fully agree that incorporating a trainable pruning model is a valuable extension and is fully compatible with the proposed PARoG. Learned pruning is orthogonal and does not change our main contribution, and we plan to explore it in the future work.
>
> ---
> ### **Weakness 3 & Question 1. Initial answers are more likely to be incorrect, and the self-refinement strategy could then introduce additional noise. Could the authors provide statistics on the proportion of initial answers that are correct?**
>
> Response to W3 & Q1. We thank the reviewer for pointing this out. To better understand how the proposed strategy refines the initial answers, we conduct additional experiments to quantify (i) how often initial answers are correct, and (ii) how effectively the refinement stage corrects initially wrong predictions.
>
> **Table i: Correctness Rate of Initial Answers.**
>
> | **Dataset** | **WebQSP** | **GrailQA** | **CWQ** |
> | --- | --- | --- | --- |
> | Correctness rate of initial answers | 61.7 | 28.9 | 43.4 |
>
> These results show that initial answers are far from perfect, especially on GrailQA and CWQ, where the correctness is below 45%. This result indicates that PARoG does not rely on strong memorization or domain-specific prior knowledge. Instead, initial answers provide only a coarse signal.
>
> **Table ii: Correct Refinement Rate on Initially Incorrect Answers.**
>
> | **Dataset** | **WebQSP** | **GrailQA** | **CWQ** |
> | --- | --- | --- | --- |
> | Correct refinement rate | 73.4 | 77.1 | 62.4 |
>
> These results demonstrate the effectiveness of the refinement mechanism. The high correction rates show that PARoG uses KG evidence to fix initial mistakes rather than amplify them. Moreover, Table 4 in the paper (page 9) already shows that removing refinement decreases performance by 2–4 points, especially for weaker LLMs. These results indicate that refinement is robust even when priors are noisy.

---

> ### Author Response · Authors · 2025-11-26
>
> ---
> ### **Weakness 4. The writing lacks clarity.**
>
> Response to Weakness 4. We thank the reviewer for this suggestion. We have revised our paper, including fixing Figure 1, adding a procedure algorithm summarization of the plan-answer-refinement paradigm, expand the experimental details, and providing more failure case studies. For better clarification, we list the model used in each stage.
>
> | **Stage** | **Model** |
> | --- | --- |
> | Planning | Fine-tuned LLaMA-3.1 |
> | Initial answer generation | GPT-3.5 / GPT-4 |
> | Exploration scoring | Same GPT-3.5 / GPT-4 |
> | Refinement | Same GPT-3.5 / GPT-4 |
>
> ---
> ### **Question 4. Were pruning stages replaced when the planner was replaced?**
>
> Response to Question 4. No. The exploration/pruning stage remained fixed. Only the planner model was swapped to isolate the effect of planning.
>
> ---
> ### **Question 5. Could the authors provide more detailed error analysis?**
>
> Response to Q5. We present three error types (Relation selection errors, Missing-entity retrieval, KG incompleteness / no valid execution path) and analyze how these errors impact the final result in **Appendix J** in the revised version of the paper.
>
> ---
> ### **Question 6. How does PARoG fundamentally differ from Plan-on-Graph (PoG)?**
>
> Response to Q6. We thank the reviewer for raising this important question. To provide a clear and structured comparison, we include the following table to summarize the fundamental differences.
>
> | **Aspect** | **Plan-on-Graph (PoG)** | **PARoG (ours)** |
> | --- | --- | --- |
> | Planning supervision | No explicit structured supervision | SPARQL-grounded decomposition |
> | Plan structure | Adaptive but mostly linear | Compositional, operator-aware, SPARQL-consistent |
> | Reasoning paradigm | KG retrieval → answer | Answer → KG retrieval → refinement |
> | Error correction | Local reflection | Explicit error-override using triples |
>
> Explanations：
>
> *   **SPARQL-grounded supervision vs. heuristic decomposition.** PoG decomposes questions heuristically using the LLM’s internal heuristics and performs sequential entity–relation expansions, which inherently induces search space truncation bias. In contrast, PARoG is trained using ground-truth SPARQL logical forms, enabling explicit handling of conjunctions, compositions, comparatives, and superlatives. This gives our planner stronger alignment with the semantics of actual query execution. **Table 9 (Appendix F)** shows the comparison between PoG and PARoG on incorrect truncation, demonstrating the effectiveness of the proposed method.
>
> *   **Different reasoning paradigms.** PoG follows a retrieve-then-answer paradigm. Conversely, PARoG instead adopts an answer-retrieval-refinement pipeline, which enables PARoG to directly mitigate error amplification, especially when initial retrieval is noisy.
>
> *   **Explicit error correction using KG triples.** PoG performs localized reflection but remains heavily dependent on the retrieved paths. Instead, PARoG explicitly verifies candidate answers using KG triples and overrides incorrect LLM predictions during refinement. This yields stronger correction ability, as proved by our new correct refinement rate statistics in **Table 6**.
>
>
> The differences between PARoG and PoG are illustrated and discussed in **Figure 1** and **Section 3.1&3.2**. We will strengthen the discussion in the final version of the paper. In summary, the core distinction is that PARoG is SPARQL-supervised and semantics-grounded, whereas PoG is heuristic and ungrounded, leading to substantially different decomposition quality and refinement behavior. These differences lead to the substantial gains observed in main comparison (**Table 1-3**, **Figure 4**) and ablation study (**Table 6** and **Table 9**).

---

> > ### Comment · Reviewer_HE4u · 2025-11-27
> >
> > Thank you for your detailed response. You have addressed the majority of my concerns, and I appreciate the additional experiments you conducted. Although I still feel that some of the improvements over PoG are incremental, the thoroughness of your experiments and clarifications has convinced me to raise my score to 6. Thanks.

---

### Official Review · Reviewer_XrmY · 2025-10-31

**Soundness:** 2
**Presentation:** 2
**Contribution:** 2
**Rating:** 4
**Confidence:** 5

**Summary:**

This paper proposes a novel framework for LLM reasoning on knowledge graphs, termed plan-answer-refine-on-graph (PARoG). To be specific, the proposed method leverages SPARQL queries from KG data as references, decomposing them into structured step-by-step plans. Then, PARoG trains LLMs to construct such structured plans for retrieval on KGs. In addition, a plan-answer-refine paradigm is adopted by first answering each sub-query and then refining the predictions. Extensive experiments demonstrate the effectiveness of the proposed method.

**Strengths:**

1.	This paper is well-organized and easy to follow.
2.	This paper provides the source code to ensure the reproducibility of the proposed method.
3.	Extensive experiments show the effectiveness of the proposed method.

**Weaknesses:**

1.	The figures could be further refined to enhance readability. In particular, the font size in Figures 1 and Figure 2 may be quite small.
2.	The paper may lack some baseline methods for comparison, such as GNN-RAG [1], SubgraphRAG [2].
3.	The core idea may be similar to PoG, as both methods adopt a comparable pipeline for answer generation (i.e., plan–answer–refine). What is the key difference between the proposed method and PoG?
4.	As stated in the exploration section, the KG exploration process in the proposed method appears similar to existing approaches. How does the proposed method address the associated challenges?

[1] Mavromatis, Costas, and George Karypis. "Gnn-rag: Graph neural retrieval for large language model reasoning." arXiv preprint arXiv:2405.20139 (2024).

[2] Li, Mufei, Siqi Miao, and Pan Li. "Simple is Effective: The Roles of Graphs and Large Language Models in Knowledge-Graph-Based Retrieval-Augmented Generation." The Thirteenth International Conference on Learning Representations.

**Questions:**

Please see in Section Weaknesses

---

> ### Author Response · Authors · 2025-11-26
>
> Thank you for your insightful comments and suggestions. We appreciate the opportunity to address your concerns and questions.
>
> ---
> ### **Weakness 1. The figures could be further refined to enhance readability.**
>
> Response to W1. Thank you for pointing this out. We have updated Figure 1 & 2 and enlarged the font size in the revised version.
>
> ---
> ### **Weakness 2. The paper may lack some baseline methods for comparison, such as GNN-RAG and SubgraphRAG.**
>
> Response to W2. We appreciate the suggestion. We have incorporated the comparison results in **Table 1** of the revised version. Both baselines are categorized as "Graph-Retrieval Methods". For readability and clarity, we present part of the comparison results directly in this response.
>
> **Table i. Performance comparison.**
>
> | **Method** | **WebQSP** | **CWQ** |
> | --- | --- | --- |
> | GNN-RAG | 82.8 | 62.8 |
> | SubGraphRag | 87.1 | 54.9 |
> | **PARoG** | **91.2** | **79.3** |
>
> Across both benchmarks, our propose method consistently outperforms the compared methods.
>
> ---
> ### **Weakness 3. The key difference between the proposed method and PoG?**
>
> Response to W3. Thank you for raising this important point. While both PoG and PARoG adopt the general idea of decomposing a question into sub-goals, the core mechanisms are fundamentally different. To provide a clear and structured comparison, we include the following table to summarize the fundamental differences.
>
> | **Aspect** | **Plan-on-Graph (PoG)** | **PARoG (ours)** |
> | --- | --- | --- |
> | Planning supervision | No explicit structured supervision | SPARQL-grounded decomposition |
> | Plan structure | Adaptive but mostly linear | Compositional, operator-aware, SPARQL-consistent |
> | Reasoning paradigm | KG retrieval → answer | Answer → KG retrieval → refinement |
> | Error correction | Local reflection | Explicit error-override using triples |
>
> Explanations：
>
> *   **SPARQL-grounded supervision vs. heuristic decomposition.** PoG decomposes questions heuristically using the LLM’s internal heuristics and performs sequential entity–relation expansions, which inherently induces search space truncation bias. In contrast, PARoG is trained using ground-truth SPARQL logical forms, enabling explicit handling of conjunctions, compositions, comparatives, and superlatives. This gives our planner stronger alignment with the semantics of actual query execution. **Table 9 (Appendix F)** shows the comparison between PoG and PARoG on incorrect truncation, demonstrating the effectiveness of the proposed method.
>
> *   **Different reasoning paradigms.** PoG follows a retrieve-then-answer paradigm. Conversely, PARoG instead adopts an answer-retrieval-refinement pipeline, which enables PARoG to directly mitigate error amplification, especially when initial retrieval is noisy.
>
> *   **Explicit error correction using KG triples.** PoG performs localized reflection but remains heavily dependent on the retrieved paths. Instead, PARoG explicitly verifies candidate answers using KG triples and overrides incorrect LLM predictions during refinement. This yields stronger correction ability, as proved by our new correct refinement rate statistics in **Table 6**.
>
>
> The differences between PARoG and PoG are illustrated and discussed in **Figure 1** and **Section 3.1&3.2**. We will strengthen the discussion in the final version of the paper. In summary, the core distinction is that PARoG is SPARQL-supervised and semantics-grounded, whereas PoG is heuristic and ungrounded, leading to substantially different decomposition quality and refinement behavior. These differences lead to the substantial gains observed in main comparison (**Table 1-3**, **Figure 4**) and ablation study (**Table 6** and **Table 9**).
>
> ---
> ### **Weakness 4. The KG exploration process in the proposed method appears similar to existing approaches. How does the proposed method address the associated challenges?**
>
> Response to W4. We agree that the low-level entity–relation expansion is similar to prior approaches as we stated in section 3.2. However, the technical contribution of PARoG is to change what is explored and how errors propagate, via two mechanisms:
>
> *   **Structured SPARQL-guided plans change what is explored.** Exploration is no longer driven by a linear path, but by compositional sub-objectives produced by the structured planner, which preserves correct candidate sets rather than pruning them prematurely. **Table 9** in **Appendix F** provides direct evidence proving the effectiveness.
>
> *   **The answer-refinement mechanism counteracts faulty retrieval.** After each exploration iteration, PARoG explicitly checks whether retrieved evidence aligns with the question and revises the tentative answer accordingly. This claim is supported by the correctness rate analysis shown in **Table 6**.
>
> Thus, while the primitive search operator is similar, the global reasoning dynamics are fundamentally different. We provide the pseudocode in Appendix C for better clarification.

---

### Official Review · Reviewer_M5aj · 2025-11-03

**Soundness:** 3
**Presentation:** 3
**Contribution:** 3
**Rating:** 6
**Confidence:** 3

**Summary:**

The paper proposes **Plan-Answer-Refine-on-Graph (PARoG)**, a KG-augmented LLM framework designed to address two limitations of prior KGQA agents: **(i)** *search-space truncation bias* from linear, top-k-pruned walks and **(ii)** *error amplification* from over-reliance on retrieved but partially relevant entities. PARoG has two pillars. First, **SPARQL-guided structured planning**: the authors decompose SPARQL queries into sub-objectives (conjunction, composition, comparatives, superlatives) and train a relatively small planner (Llama-3.1-8B) via SFT to emit stepwise plans with uniform granularity. Second, a **Plan→Answer→Refine** loop: for each sub-objective, the model first proposes a tentative answer using parametric knowledge, then executes KG exploration and **self-refines** the answer against retrieved triples, explicitly checking alignment and stopping when sufficient. Experiments on **WebQSP, CWQ, and GrailQA** report SOTA Hits@1 with fewer LLM calls and tokens than ToG/PoG; ablations show (a) self-refinement materially helps and (b) the 8B SPARQL-supervised planner outperforms much larger planning LLMs (GPT-3.5, DeepSeek-R1).

**Strengths:**

* **Addresses core failure modes with targeted design.** SPARQL-guided planning avoids linear-path pruning; Answer→Refine reduces retrieval-induced error amplification.
* **Solid, multi-benchmark gains with efficiency.** PARoG improves Hits@1 over ToG/PoG/KG-Agent on WebQSP, CWQ, and GrailQA, with fewer calls and tokens.
* **Decision-useful ablations.** Self-refinement helps markedly (esp. with GPT-3.5); the SPARQL-trained 8B planner outperforming larger LLMs is practically important.
* **Clear problem decomposition.** The taxonomy of sub-query types (conjunction/composition/comparatives/superlatives) aligns well with KGQA structure.

**Weaknesses:**

1. **Statistical rigor.** Main results lack **CIs/multi-seed variance**; given RL-like exploration and LLM stochasticity, uncertainty reporting is important.
2. **Planner data generation & potential biases.** The **SPARQL→plan** dataset (74,802 examples via GPT-4o) is central, but quality controls, inter-annotator checks, and error taxonomies are not reported; robustness to noisy/ambiguous SPARQL is unclear.
3. **Faithfulness audits.** The refinement step relies on LLM judgments of alignment; the paper does not quantify hallucinated/irrelevant triples that slip through nor enforce execution-based correctness constraints during refinement.
4. **Generality beyond Freebase.** No experiments on **Wikidata/DBpedia** or text-augmented regimes; claims of broad applicability would be stronger with heterogeneous KGs or KG incompleteness stress tests.
5. **Missing engineering details.** Exact prompts, stopping criteria, and filtering thresholds for relation/entity selection are described at a high level; reproducibility would benefit from fuller protocol disclosure.

**Questions:**

1. **Statistics & stability.** Please report **means ± 95% CI over ≥3 seeds** for all main tables and ablations; include sensitivity to planner size and decoding parameters.
2. **SPARQL→plan pipeline.** How do you validate the GPT-4o decompositions for **semantic equivalence** to the original SPARQL, and what is the estimated error rate? Any manual spot-checks?
3. **Execution-faithfulness checks.** During refinement, can you enforce **KG-execution constraints** (e.g., verifying that asserted triples exist; type constraints) and report the rate at which refinement overturns an initially wrong tentative answer?
4. **Generality.** Do results transfer to **Wikidata** (different schema and aliasing) and to **KG+text** settings (e.g., Freebase+Wikipedia)? If not, what components would need adaptation?
5. **Efficiency trade-offs.** Table 5 shows fewer calls and tokens; could you add **cost-accuracy curves** (vary beam/iterations) and detail caching/parallelization assumptions used for accounting?
6. **Error analysis.** Appendix B indicates high failure due to missing answer entities in retrieval for prior methods; can you provide a comparable breakdown for PARoG and the fraction resolved by the refine stage?

---

> ### Author Response · Authors · 2025-11-26
>
> We thank the reviewer for the constructive and detailed feedback. We address all concerns below.
>
> ---
> ### **Weakness 1 & Question 1. Statistical rigor (CIs and multi-seed stability)**
>
> Response to W1 & Q1. We agree that reporting uncertainty is essential for LLM-based exploration. We have rerun all main experiments and ablations under 3 random seeds and now report mean ± 95% CI for all settings. We have updated **Table 1, 2** and **Appendix G** in the revised paper to include these results.
> The results are also listed in this response below.
>
>
> **Table i. Main Results with ChatGPT-3.5**
>
> | **Model** | **WebQSP** | **GrailQA** | **CWQ** |
> | --- | --- | --- | --- |
> | ToG | 76.4 ± 1.9 | 68.9 ± 1.5 | 57.2 ± 1.8 |
> | PoG | 82.1 ± 2.2 | 76.5 ± 2.1 | 63.2 ± 1.2 |
> | PArog | **89.0 ± 1.3** | **82.7 ± 1.5** | **73.1 ± 0.9** |
>
> **Table ii. Main Results with GPT-4**
>
> | **Model** | **WebQSP** | **GrailQA** | **CWQ** |
> | --- | --- | --- | --- |
> | ToG | 80.7 ± 1.7 | 80.9 ± 1.4 | 65.6 ± 2.2 |
> | PoG | 87.3 ± 1.5 | 84.3 ± 1.8 | 75.0 ± 1.4 |
> | PArog | **91.2 ± 0.9** | **87.1 ± 1.3** | **79.3 ± 1.1** |
> ---
> PARoG consistently improves over the strongest baselines with both higher means and smaller confidence intervals, indicating a more stable reasoning process.
>
> **Table iii. Ablation study on the planner module.**
>
> | **Method** | **WebQSP** | **GrailQA** | **CWQ** |
> | --- | --- | --- | --- |
> | GPT-3.5 | 83.1 ± 1.4 | 76.7 ± 1.6 | 65.4 ± 1.2 |
> | Deepseek-R1 | 88.2 ± 1.5 | 80.2 ± 1.3 | 68.7 ± 1.1 |
> | llama3.1-8B (ours) | 89.0 ± 1.3 | 82.7 ± 1.5 | 73.1 ± 0.9 |
> | llama3.2-1B (ours) | 87.2 ± 1.2 | 79.1± 1.6 | 68.5±1.1 |
> | llama2-13B (ours) | 88.4± 1.4 | 81.9± 1.2 | 70.3± 1.0 |
>
> It can be observed that our smaller planners outperforms larger general LLMs. Even the smallest 1B planner still achieves comparable performance compared to Deepseek-R1, confirming that SPARQL-guided training provides a stronger planning signal than scaling model size alone.
>
> **Table iv. Ablation study on the self-refinement mechanism with ChatGPT-3.5.**
>
> | Method | WebQSP | GrailQA | CWQ |
> | --- | --- | --- | --- |
> | w/SR | 89.0 ± 1.3 | 82.7 ± 1.5 | 73.1 ± 0.9 |
> | w/o SR | 88.0 ± 1.6 | 78.9 ± 1.4 | 69.2 ± 1.2 |
>
> **Table v. Ablation study on the self-refinement mechanism with ChatGPT-4.**
>
> | Method | WebQSP | GrailQA | CWQ |
> | --- | --- | --- | --- |
> | w/SR | **91.2 ± 0.9** | **87.1 ± 1.3** | **79.3 ± 1.1** |
> | w/o SR | 89.7 ± 1.1 | 85.5 ± 1.4 | 77.2 ± 1.0 |
>
> Self-refinement consistently improves both the mean accuracy and the variance, validating its role in mitigating error amplification under both weaker and stronger base LLMs.
>
> To conclude, we now report 3-seed mean ± 95% CI for all main results. PARoG shows higher accuracy and consistently smaller confidence intervals than all baselines. The results confirm the stability and reproducibility of the SPARQL-guided planner and the Answer-Refine mechanism.
>
> ---
> ### **Weakness 2 & Question 2. SPARQL→Plan data quality, error rate, and manual spot-checks.**
>
> Response to W2 & Q2. We thank the reviewer for raising this important point. The responses are as follows.
>
> *   Semantic consistency is structurally enforced by our SPARQL-supervised mapping process. As described in Section 3.1 "Semantic Consistency Mapping", our dataset construction first decomposes the SPARQL program into atomic operations, and then uses GPT4 to reconstruct the corresponding natural-language query from these operations.  Because the natural-language sub-questions are generated after recovering the SPARQL structure, the semantic alignment between the sub-objectives and the original query is inherently preserved. This mapping procedure already guarantees a strong form of semantic consistency.
>
> *   The entire dataset is generated automatically without human intervention. As shown in **Appendix L.1** in the revised paper, we provide the full prompts used to produce the decomposed plans and rephrased natural-language queries. The whole pipeline is fully automated without any manual annotation or post-editing, which ensures reproducibility and scalability.
>
> *   We plan to include a quantitative quality assessment (e.g., semantic equivalence rate and error taxonomy) in the final version.

---

> ### Author Response · Authors · 2025-11-26
>
> ---
> ### **Weakness 3 & Question 3. Execution-faithfulness checks.**
>
> Response to W3 & Q3. We thank the reviewer for raising this important point on the faithfulness of the refinement step. We have conducted additional analyses to quantify (i) how often the tentative answers produced by the LLM are correct, and (ii) how effectively the refinement stage corrects initially wrong answers using KG evidence.
>
>
> **Table i: Correctness Rate of Initial Answers.**
>
> | **Dataset** | **WebQSP** | **GrailQA** | **CWQ** |
> | --- | --- | --- | --- |
> | Correctness rate of initial answers (%) | 61.7 | 28.9 | 43.4 |
>
> These numbers verify that LLM parametric knowledge is often incomplete or unreliable, highlighting the necessity of a refinement stage grounded in KG execution.
>
> **Table ii: Correct Refinement Rate on Initially Incorrect Answers.**
>
> | **Dataset** | **WebQSP** | **GrailQA** | **CWQ** |
> | --- | --- | --- | --- |
> | Correct refinement rate (%) | 73.4 | 77.1 | 62.4 |
>
> These results show that the refinement stage successfully overturns a large proportion of incorrect tentative answers.
>
> As described in Sec 3.2, the model is required to verify whether the retrieved KG evidence aligns with the sub-objective and the question at each exploration step. If the evidence contradicts the tentative answer, the refinement module replaces the original prediction accordingly.
> Thus, refinement is explicitly grounded in KG execution, not in free-form re-generation or hallucinated reasoning.
> These results have been included in the revised version of the paper (**Table 6** and **Appendix F**).
>
> ---
> ### **Weakness 4 & Question 4. Generality beyond Freebase.**
>
> Response to W4 & Q4. We thank the reviewer for raising the question of whether PARoG generalizes to other knowledge graphs beyond Freebase. To address this, we have added a new generalization study comparing PARoG, PoG, and ToG on using **Wikidata** as the underlying KG. These results are now included in **Table 4** of the revised paper.
>
> **Table i. Generalization Study：performance comparison using different source KGs.**
>
> | **Method** | **WebQSP** | **CWQ** |
> | --- | --- | --- |
> | _with Freebase_ |  |  |
> | ToG | 76.2 | 57.1 |
> | PoG | 82.0 | 63.2 |
> | **PARoG** | **89.3** | **73.3** |
> | _with WikiData_ |  |  |
> | ToG | 68.6 | 54.9 |
> | PoG | 73.8 | 60.7 |
> | **PARoG** | **79.1** | **69.5** |
>
> Notably, the absolute performance on Wikidata is lower than that on Freebase because the datasets are originally annotated for Freebase. Moreover, Wikidata is substantially larger and more heterogeneous, which increases the difficulty for KG exploration and relation filtering.
>
> Compared with the baselines, PARoG consistently produces substantial improvements over compared approaches under the WikiData setting. The results demonstrate that the proposed planner and answer-refinement mechanism are not tied to specific relations but generalize well under different schema organizations, relation granularities, and naming conventions.
>
> ---
> ### **Weakness 5. Engineering details.**
>
> Response to W5. We have added the engineering details in the revised version of paper (**Appendix L**), including full prompts , the SPARQL templates, and stopping criteria. It should be noted that, as described in Appendix L, no threshold is required in our work.
>
> ---
> ### **Question 5. Efficiency trade-offs.**
>
> Response to Q5. We thank the reviewer for requesting a more detailed analysis of the efficiency–accuracy trade-off. To address this, we provide iteration–accuracy curves and iteration–cost (LLM calls) curves, which are now included in **Appendix G** of the revised version. These results quantify how PARoG scales with the number of refinement iterations.
>
> **Table i: Iteration-accuracy curve:**
>
> | **Iteration** | **1** | **2** | **3** | **4** | **5** |
> | --- | --- | --- | --- | --- | --- |
> | **Accuracy (Hit@1)** | 61.3 | 64.3 | 69.4 | 73.1 | 74.2 |
>
> **Table ii: Iteration-cost (LLM calls) curve:**
>
> | **Iteration** | **1** | **2** | **3** | 4 | 5 |
> | --- | --- | --- | --- | --- | --- |
> | **LLM calls** | 3.1 | 5.5 | 7.9 | 10.2 | 12.1 |
>
>
> ---
> ### **Question 6. Error analysis.**
>
> Response to Q6. We have presented three error types (Relation selection errors, Missing-entity retrieval, KG incompleteness / no valid execution path) and analyzed how these errors impact the final result in **Appendix J** in the revised version of the paper.

---

### Author Response · Authors · 2025-11-26
**Response to All Reviewers.**

We sincerely thank all reviewers for their insightful feedback. Following the suggestions, we have made substantial revisions to the paper, including:
- **Presentation improvements**, such as fixing Figure 1, enlarging the font size in figures, streamlining Section 3, providing pseudocode for the answer–refinement exploration, and adding missing references.
- **New generalization experiments on Wikidata (Table 3)**, demonstrating the transferability of PARoG beyond Freebase KGs.
- **Stability analysis (Appendix G)**, evaluating variance across multiple seeds.
- **Planner parameter–scaling analysis (Appendix G)**, comparing planners of different model sizes.
- **Search-space truncation bias analysis (Appendix F)**, quantifying the reduction in truncation errors.
- **Efficiency trade-off (Appendix H)**, characterizing the trade-off between efficiency and accuracy.
- **Initial-answer correctness analysis (Appendix F)**, measuring the accuracy of LLM tentative answers before refinement.
- **Effectiveness of the Self-Refinement mechanism (Table 6)**, added analysis showing the percentage of incorrect initial answers corrected by the refinement step, demonstrating its essential role.
- **Expanded failure-case analysis (Appendix J)**, covering error sources such as conflicting KG facts and deep compositional reasoning failures.
- **Full details of LLM prompt templates and scripts (Appendix L)**.

We hope these revisions sufficiently address the concerns raised and look forward to the reviewers’ further comments.

---

### Author Response · Authors · 2025-12-04
**Global Response**

To the Area Chairs:

We would first like to sincerely thank the Area Chair for carefully coordinating the review and discussion process. We also thank the four reviewers (M5aj, XrmY, HE4u, iDwB) for their constructive comments. During the rebuttal period, we provided detailed responses to all questions and updated the paper accordingly. To help the Area Chair quickly grasp the key points of the discussion, we summarize below the major revisions and the main feedback from the reviewers:
## Summary of Revisions:
Addressing the reviewers' core concerns, we have made substantial revisions to the paper.
- Presentation improvements, such as fixing Figure 1, enlarging the font size in figures, streamlining Section 3, providing pseudocode for the answer–refinement exploration, and adding missing references.
- New generalization experiments on Wikidata (Table 3), demonstrating the transferability of PARoG beyond Freebase KGs.
- Stability analysis (Appendix G), evaluating variance across multiple seeds.
- Planner parameter–scaling analysis (Appendix G), comparing planners of different model sizes.
- Search-space truncation bias analysis (Appendix F), quantifying the reduction in truncation errors.
- Efficiency trade-off (Appendix H), characterizing the trade-off between efficiency and accuracy.
- Initial-answer correctness analysis (Appendix F), measuring the accuracy of LLM tentative answers before refinement.
- Effectiveness of the Self-Refinement mechanism (Table 6), added analysis showing the percentage of incorrect initial answers corrected by the refinement step, demonstrating its essential role.
- Expanded failure-case analysis (Appendix J), covering error sources such as conflicting KG facts and deep compositional reasoning failures.
- Full details of LLM prompt templates and scripts (Appendix L)
## Status of Reviewer Responses:
Reviewer iDwB (Initial Score: 6; Post-rebuttal: 8): After we supplemented the paper with the requested experiments, the reviewer was satisfied with our response and decided to raise the score to 8.

Reviewer HE4u (Initial Score: 4; Post-rebuttal: 6): The reviewer thanked us for our detailed response, noted that we had addressed most of the concerns, and appreciated the additional experiments. The reviewer further stated that our thorough experiments and clarifications were sufficient to raise the score to 6.

We thank all reviewers again for their hard work and valuable suggestions.

---

### Meta-Review · Area_Chair_ac3G · 2026-01-07

**Summary:**

This paper identifies two limitations in existing KG-augmented LLMs, namely (1) search space truncation bias, and (2) entity error amplification. To address these issues, the authors propose Plan-Answer-Refine-on-Graph (PARoG), a framework that can decompose the queries into structured plans. It then leverages LLM to answer each sub-query independently, and refine its prediction by integrating evidence retrieved from the KG. Extensive experiments are conducted on multiple KG reasoning benchmarks to demonstrate the effect of the proposed framework.

Overall, this paper is well motivated by precisely pointing out two existing limitations. The proposed framework is clearly tailored towards closing these gaps. The Plan-Answer-Refine pipeline effectively avoids search space truncation issue and knowledge conflict between LLMs and KGs. Experiments on multiple benchmarks also demonstrate clear gains over existing methods, both in performance and efficiency, with informative ablations.

Meanwhile, reviewers agree on the following weaknesses which require further revision:
- Technical contribution over existing baselines such as PoG: The pipeline appears similar to existing works. While the authors explain that each stage has its own merits such as supervised training to produce structured plan and answer-then-refine compared with retrieve-then-answer, these changes do not hold significant technical novelty, or at least deserve further clarifications.
- As the framework relies on training of a good planner, the quality of the generated plan data is unknown. Manual checks and error statistics are recommended to demonstrate the reliability.
- Reliance on domain-specific datasets when training the planner, limiting scalability

I appreciate the authors' extensive efforts in conducting additional experiments to address the reviewers' concerns. I agree that many concerns have been addressed, but the technical contribution compared with existing frameworks deserves further discussions.

**Reviewer Concerns:**

Concerns being addressed:
- Statistical rigor: multi-run with mean and variance reporting has been suggested and implemented by the authors during rebuttal.
- Faithfulness of the refinement step: the authors have run additional experiments demonstrating the original correctness rate of using LLMs versus the correction rate after the refinement.
- Generalization towards other KG domains beyond freebase is verified.
- Engineering details, error case analysis and efficiency trade-offs are supplemented.
- Additional baselines suggested by reviewers are added.
- More proofs that PARoG mitigates truncation bias using additional experiments

Outstanding concerns:
- The lack of quality checks for plan data generation. Manual checks and error statistics are recommended.
- Technical contribution over existing baselines such as PoG: The pipeline appears similar to existing works. While the authors explain that each stage has its own merits such as supervised training to produce structured plan and answer-then-refine compared with retrieve-then-answer, these changes do not hold significant technical novelty, or at least deserve further clarifications.
- Reliance on domain-specific datasets when training the planner, limiting scalability

**Reviewer Scores:**

Two reviewers have increased scores after rebuttal. Another two scores remain the same, and I don't think there will be further changes to the score.

---

### Decision · Program_Chairs · 2026-01-26

Accept (Poster)